# Contextualize-then-Aggregate: Circuits for In-Context Learning in Gemma-2 2B

**Aleksandra Bakalova, Yana Veitsman, Xinting Huang & Michael Hahn**
Saarland Informatics Campus, Saarland University
{abakalov, yanav, xhuang, mhahn}@lst.uni-saarland.de

## Abstract

In-Context Learning (ICL) is an intriguing ability of large language models (LLMs). Despite a substantial amount of work on its behavioral aspects and how it emerges in miniature setups, it remains unclear which mechanism assembles task information from the individual examples in a few-shot prompt. We use causal interventions to identify information flow in Gemma-2 2B for five naturalistic ICL tasks. We find that the model infers task information using a two-step strategy we call *contextualize-then-aggregate*: In the lower layers, the model builds up representations of individual few-shot examples, which are contextualized by preceding examples through connections between few-shot input and output tokens across the sequence. In the higher layers, these representations are aggregated to identify the task and prepare prediction of the next output. The importance of the contextualization step differs between tasks, and it may become more important in the presence of ambiguous examples. Overall, by providing rigorous causal analysis, our results shed light on the mechanisms through which ICL happens in language models. [1]

## 1 Introduction

In-Context Learning (ICL) is an intriguing property of large language models and has spurred a substantial amount of interest into how transformers are able to perform it (e.g. Brown et al., 2020; Min et al., 2022; Garg et al., 2022; Akyürek et al., 2023; Cho et al., 2025; Wang et al., 2023). Prior work in mechanistic interpretability has found function vectors (Todd et al., 2024; Hendel et al., 2023): attention heads whose output encodes task information, and which are causally responsible for the prediction of the response. However, the circuit by which these vectors are assembled remains only partly understood. Recent work has proposed that each few-shot example's output token computes task information, which is then aggregated by attention heads to predict the next output (Wang et al., 2023; Cho et al., 2025; Kharlapenko et al., 2025), but this strategy has also been observed to leave part of the models' performance unexplained (Cho et al., 2025).

In this paper, we use causal interventions (e.g. Vig et al., 2020; Geiger et al., 2021; Meng et al., 2022) to identify a circuit performing ICL on five naturalistic tasks formatted as few-shot prompts (Capitalization, Country-Capital, Present-Past, Person-Sport, Copying) in Gemma-2 2B (Team et al., 2024). We replicate the relevance of the aggregation step suggested by prior work, but show that it explains only a portion of the model's full performance. We use causal interventions to identify information flow that recovers at least 90% of the model's performance. A key idea is to first identify a computation graph between the tokens in a prompt (Figure 1). We find that the required graph differs between tasks: whereas the simplest circuit is largely sufficient on some tasks, others require additional computation paths. We find that the representations of few-shot examples later in the prompt are contextualized by information from prior few-shot examples, particularly, the last preceding few-shot example. Using causal interventions, we show that contextualization transports information about the input and output spaces, and the task itself. In particular,

---

[1] https://github.com/lacoco-lab/icl_circuits

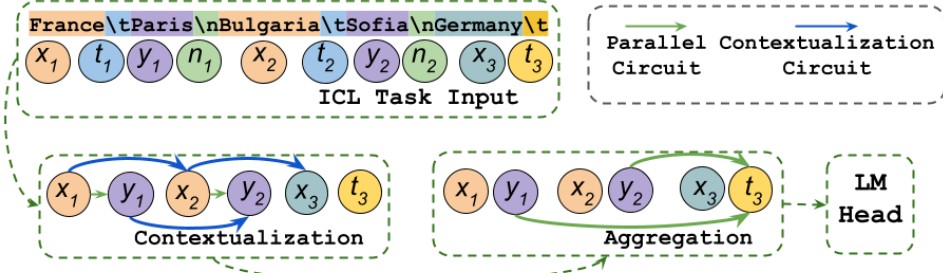

Figure 1: An $N$-shot ICL prompt (here, $N = 2$), at the example of the Country-Capital task. The green edges describe a **AGGREGATION subcircuit** that assembles information at each few-shot example in parallel ($x_i \rightarrow y_i$) and then **aggregates** these at $t_{N+1}$. The blue edges describe a **CONTEXTUALIZATION subcircuit** that contextualizes the representation of each few-shot. A circuit involving both components recovers most of Gemma2-2B's performance; aggregating without contextualization leads to a breakdown in performance on this task. See Figure 24 for other tasks.

- Aggregation edges generally move information about the task from individual examples to the last token. In the Person-Sport task, they move information about input/output spaces without regard to functional relationship.
- Contextualization edges generally move information about input/output spaces, and sometimes the task.

Overall, our main contributions are to:

1. *Obtain causally faithful information flow from the few-shot examples to the next-token prediction, explaining $\geq 90\%$ of the model's performance.*
   A key idea here is to focus on *position-level circuits*, where information flows only between a restricted subset of positions in a prompt, which is key to making the resulting computation graph interpretable.

2. *Establish the importance and function of a contextualization step that precedes aggregation.*
   This is distinct from prior proposals (Cho et al., 2025; Kharlapenko et al., 2025) focusing on the aggregation step. Contextualization becomes even more important in the presence of ambiguity (Section 3.3). Indeed, it is beneficial even in a synthetic setup (Appendix A).

## 2 Background: Aggregation and Function Vectors

Prior work on the mechanics of ICL in LLMs provides a few robust insights. Multiple studies document the existence of *task vectors* or *function vectors* (Hendel et al., 2023; Todd et al., 2024; Song et al., 2025; Kharlapenko et al., 2025; Yin and Steinhardt, 2025): Several attention heads at the last prompt token (in our case, $t_{N+1}$) output vectors that, across ICL tasks, encode task information. Erasing them prevents ICL; patching them elsewhere leads to execution of the task. Todd et al. (2024) specifically operationalize Function Vector Heads as heads whose activations, when patched from valid ICL prompts to shuffled uninformative prompts, can most increase the likelihood of the desired target. Yin and Steinhardt (2025) find that Function Vector Heads are more important to ICL than Induction Heads, hypothesized to be key in prior work (Olsson et al., 2022; Crosbie and Shutova, 2024) (cf. also Bansal et al. (2023)). However, it remains largely open what circuit assembles this information, and how information is assembled from the different few-shot examples to these function vectors.

Kharlapenko et al. (2025) discover a sparse feature circuit in Gemma-2 2B, using sparse autoencoder (SAE) features encoding task information (in few-shot examples) and features that execute them (at $t_{N+1}$). However, the circuit does not distinguish between inputs and outputs for different few-shot examples, leaving open whether task information is

created independently at each example or whether the representations are dependent across few-shot examples. Cho et al. (2025) suggest a different circuit: (1) assembling an encoding of each few-shot example's input and moving it to few-shot label position, (2) attending to labels whose input is similar to the query, and copying these labels to the query. Their experiments are restricted to semantic text classification (e.g., sentiment), in which few-shot inputs typically are either semantically aligned or opposed. In addition, the arguments are largely based on evaluating representational similarity and on supervised probing, which does not establish that this mechanism is causally sufficient. While a causal intervention, through zeroing out the relevant components, shows this mechanism is relevant to ICL (Section 5.1 in Cho et al. (2025)), it leaves open if the mechanism alone can fully explain the behavior, or whether it is only a part of the overall mechanism. Overall, while function vectors play an important role for many ICL tasks, the information flow leading to aggregation of task information across few-shot examples remains unclear. To foreshadow our results, we show that, while parallel aggregation of per few-shot representations is important, the overall circuit is task-dependent, and may require a variety of additional components which, as we will show, contextualize the per few-shot representations on other few-shot examples.

## 3 Results

We focus on Gemma-2, especially its 2B version, but our methodology allows us to also investigate larger variants (9B and 27B). We evaluated the model on tasks from Todd et al. (2024), and determined five tasks on which the 2B model achieves high (between 88% and 100%) accuracy: Copying [2], Present-Past, Capitalization, Country-Capital, Person-Sport (Figure 2).[3] We decided to focus on tasks with high accuracy in order to obtain a clear signal, as for these tasks there is a clear understanding of the input-output behavior exhibited by the full model. We focused experiments on shorter (3-shot) and longer (10-shot) prompts.

Each of the five tasks has an input space $\mathcal{X}$ (e.g., arbitrary lowercase words for Capitalization; countries for Country-Capital) and an output space $\mathcal{Y}$ (e.g., arbitrary capitalized words for Capitalization; cities for Country-Capital), and defines a function $f : \mathcal{X} \rightarrow \mathcal{Y}$ (e.g., mapping countries to their capitals) (Appendix, Figure 25). Each $N$-shot prompt then has the following structure: $x_1 t_1 y_1 n_1 \dots x_N t_N y_N n_N x_{N+1} t_{N+1}$, $\forall i : x_i \in \mathcal{X}, y_i \in \mathcal{Y}, y_i = f(x_i), t_i = \backslash \text{t}, n_i = \backslash \text{n}$.

### 3.1 Identifying Circuits

To localize the behavior of the model, we use *patching*, an approach widely adopted in the literature (Wang et al., 2022; Hanna et al., 2023). With this technique, a model is viewed as a computation graph with activations in different layers as nodes, and computations between them as edges. We then *ablate* some of the edges in the graph, forcibly replacing computation along a specific edge with the one computed on the *counterfactual* input. Counterfactual inputs are designed to erase relevant information from the input while leaving other information intact for localization of model behavior. For example, when substituting "Berlin" with counterfactual "Paris", we isolate computations specific to the city's identity, not those activated by its category (city) or token-type (word). If ablating a set of edges does not lead to a drop in the model's performance, then the information unique to the original input relative to the counterfactual input, which was transferred along these edges, is not causal for the model's prediction. This allows to discover *circuits* – subgraphs of the full model's computation graph explaining a major part of model performance on a specific task. In contrast to the standard approach, we focus on the information flow between positions in a prompt; consequently, we differentiate between activations of the same heads at different positions. We keep nodes for $x_i$ and $y_i$ in different examples distinct, which allows us to analyze the information flow between few-shot examples.[4]

---

[2]This is not included in Todd et al. (2024), but we added it due to its naturalness.

[3]We also evaluated six other models at the 2B/3B scale, finding that they overall underperformed Gemma-2 2B at 10 shots on these tasks (Table 7), further motivating our focus on Gemma-2.

[4]This contrasts with Kharlapenko et al. (2025), who collapse all $x_i$ and $y_i$ nodes into one node each.

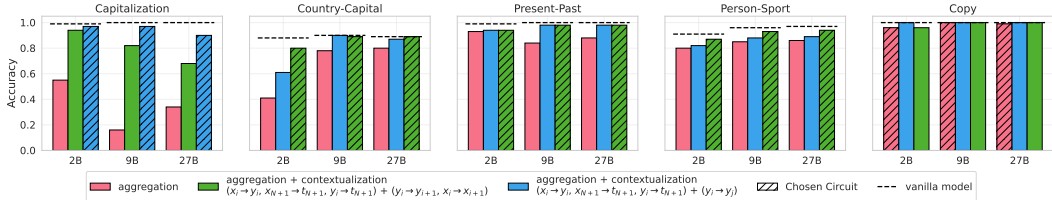

Figure 2: Accuracy of circuits (Gemma-2 2B, 9B, 27B). The AGGREGATION circuit under-performs the full model by a large margin in some tasks. Contextualizing edges recover a large part of the gap. Whereas the AGGREGATION already performs well in some tasks, we note that the contextualizing edges become crucial when evaluating on a difficult subset designed to be ambiguous between two tasks (Section 3.3).

We first discover a **position-level circuit**. Nodes are the positions in the prompt; edges are directed arcs between them, representing the flow of information (Figure 1). Importantly, in position-level circuits we collapse all edges across layers into a single edge. Ablating such an edge means ablating all corresponding edges in every attention head and layer. For illustration, $x_2$ in a position-level circuit can only receive four incoming edges: from $x_1, t_1, y_1$, or $n_1$. [5] Position-level circuits provide useful and interpretable upper bounds on information flow, though they do not distinguish between computations happening in different layers. We treated all tokens within a few-shot input $x_i$ or output $y_i$ together, as the number of tokens varies between few-shot examples. As positions, we distinguish the few-shot inputs $x_i$ and outputs $y_i$ and the separators within ($t_i$) and between ($n_i$) examples; though $n_i$ play no role in any of the identified circuits.

**Patching Methodology**  To identify circuits, we define a simple class of counterfactual inputs where all few-shot inputs and outputs (including the query $x_{N+1}$) are replaced with random words (Appendix H). The corrupted prompts still contain the same separators, and maintain the overall number of tokens of each $x_i/y_i$ (Appendix H, Figure 22). When we *ablate* an edge from position A to position B, the key (K) and value (V) activations of A when queried by B are replaced with activations computed on a corrupted prompt. The K/V activations of A when queried by other positions, as well as Q activation of B remain clean (Appendix H, Figure 28). This patching is applied simultaneously at each layer and head. We define the *output of a circuit* on an input prompt by greedily decoding a response $y_{N+1}$ until a separator is generated, while ablating all edges outside the circuit.[6]

**AGGREGATION Circuit: Computing representations at each example and aggregate** Based on prior work, we first hypothesized a minimal circuit consisting of edges between few-shot inputs and outputs ($x_i \rightarrow y_i$), from few-shot outputs to the last separator ($y_i \rightarrow t_{N+1}$), and also from the query to the last separator ($x_{N+1} \rightarrow t_{N+1}$).[7] We dub this the AGGREGATION circuit, as it allows each few-shot to assemble its own representation based on ($x_i, y_i$) in parallel, and then allows $t_{N+1}$ to make its prediction based on both the query and the set of few-shot examples (green edges in Figure 1). This is the primary mechanism suggested by prior work (Cho et al., 2025; Wang et al., 2023; Kharlapenko et al., 2025). We evaluated the degree to which this subcircuit is *sufficient* for task performance.[8] For this,

---

[5]The edges from/to bos/eos tokens are always ablated (Appendix E).

[6]We do, however, retain edges supporting the autoregressive prediction of the answer ($x_{N+1} \rightarrow y_{N+1}$ and $t_{N+1} \rightarrow y_{N+1}$) (Appendix E).

[7]We also always include all edges from one position to itself, and never ablate these Appendix E.

[8]While Cho et al. (2025) showed zeroing out these edges to hurt model performance, establishing that they do play a role, their method does not show *sufficiency* of the subcircuit. Interestingly, Cho et al. (2025) propose that $x_i$ and $y_i$ are connected via $t_i$, whereas we find a direct connection is largely sufficient, simplifying the resulting circuit. We note that Cho et al. (2025) found that performance suffers when zeroing out $x_i \rightarrow t_i$ or $t_i \rightarrow y_i$ and hypothesized that these edges transport information about $x_i$. However, they did not verify that these edges provide information about $x_i$ (rather than, e.g., about the presence of a prompt template). In contrast, our patching methodology allows us to

we ablated all edges other than those in this minimal circuit. With this ablation, the model achieves $\geq 85\%$ of the full accuracy on three tasks, but shows a huge drop ($< 60\%$ of full model) on Capitalization and Country-Capital (Figure 2). This confirms that the mechanism is causally important, but insufficient for overall explaining model behavior.

**CONTEXTUALIZATION Subcircuit: Edges between different few-shot examples**   We next investigated which edges are needed beyond the AGGREGATION subcircuit, aiming to find a small circuit recovering $\geq 90\%$ of the accuracy of the full model. We call all the edges in this circuit except those in the AGGREGATION subcircuit a CONTEXTUALIZATION subcircuit. Specifically, the CONTEXTUALIZATION subcircuit comprises edges that *contextualize* the representation of $(x_i, y_i)$ using prior few-shot examples $(x_j, y_j)$, where $j < i$. These edges connect nodes $A \rightarrow B$, with $A \in \{x_j, y_j | j < i\}$ and $B \in \{x_i, y_i\}$ for $i \leq N$. Note that the exact edges in this subcircuit may vary slightly across tasks.

We first ablated only edges involving non-final separators from the full model, and found this to do little harm across the five tasks compared to the full model, especially at 10 shots and for the 9B model (Tables 2, 5, 6), showing that relevant task information need not be routed through the separators in and between the few-shots, i.e., the separators may provide information about the presence of a prompt template, but do not play a nonredundant role in assembling information about the task. We thus continued with these edges ablated. We next considered the remaining logical possibilities, grouped by the types (input, output, separator) of positions: *within-type* edges (1) $x_i \rightarrow x_j$ (for all $1 \leq i < j \leq N + 1$), (2) $y_i \rightarrow y_j$ (for all $1 \leq i < j \leq N$), and *across-type* edges: (3) $x_i \rightarrow y_j$ ($1 \leq i < j \leq N$), (4) $x_i \rightarrow t_{N+1}$ ($1 \leq i \leq N$), (5) $y_i \rightarrow x_j$ ($1 \leq i < j \leq N + 1$). We investigated adding each of these groups individually to the aggregation circuit (Table 1). We found that adding (1) and (2) provided strong accuracy gains on some tasks, and focused on these two groups.

We next focused on different circuits involving these edges. We considered both general edge sets of types (1) and (2), and the local subset ($x_i \rightarrow x_{i+1}$ and $y_i \rightarrow y_{i+1}$); the latter were sufficient in all but one case ($y_i \rightarrow y_j$ needed for Capitalization). For each task, we chose the simplest circuit achieving $\geq 0.9$ at $N = 3$, or (if there is none) the circuit achieving highest accuracy at $N = 3, 10$, all based on the 2B model's accuracies (Tables 2). We illustrate the result for Country-Capital in Appendix, Figure 1; circuits for all five tasks are shown in Appendix, Figure 24. At 2B parameters and 10 few-shots, the circuits explained $\geq 90\%$ of the full model's performance, far exceeding the aggregation-only circuit on some tasks (Figure 2). We next evaluated the circuits at 9B and 27B, using the same methodology (Tables 5, 6). At 10 shots, the circuits that we had chosen based on the 2B model explained at least 90% of the full model's performance in these model. Importantly, the AGGREGATION circuit again underperformed the circuits involving contextualization on most tasks (Figure 2).

**Projecting to Activation-Level Circuits**   In order to identify the heads responsible for the information flow we have identified, we next projected the position-level circuits to **activation-level circuits**. Here, nodes are activations of attention heads indexed by both their positions and layers; edges are directed arcs between the input and output of individual attention heads. More precisely, heads can be defined as tuples of (layer, head, position), and edges can be described as (layer, head, start position, end position). We used a gradient-based method (Michel et al., 2019; Syed et al., 2023) (see Appendix F for technical details) to obtain a circuit on the level of activations with edges defined by attention heads, in the 3-shot setup (Figures 9–13). The circuit reveals that the contextualization step tends to precede the aggregation step: contextualization happens in the lower half layers, most aggregation edges are in the middle and upper layers. Prior work has established the existence of function vector heads, specific heads aggregating causally relevant task information at $t_{N+1}$ (Section 2). We next identified function vector heads by using the method described in Todd et al. (2024) for each task independently and selecting the top-10-scoring heads for each

---

conclude that these edges, at least in Gemma-2B on our prompt template, need not causally provide information beyond the presence of a prompt template (which is kept constant in our corrupted inputs). We caution that this difference need not invalidate the conclusion of Cho et al. (2025) as they used a different template, using the word "label" for $t_i$, whereas we use the "\t" punctuation.

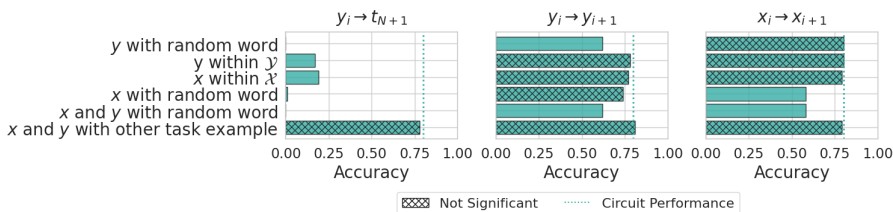

Figure 3: Patching experiments manipulating information about the input available to each edge in the full circuit, at the example of the Country-Capital task, 10-shot. For each ablation, we indicate whether the drop in performance is statistically significant compared to the full circuit performance in a binomial test with $\alpha = 0.05$. See Figure 8 for results from all tasks.

task. These heads were largely part of the identified circuits and mostly contributed to the $y_i \rightarrow t_{N+1}$ edges (Appendix, Figure 26). This confirms that the circuits we have identified describe the buildup of task information up to the function vector heads.

## 3.2  Which Information is Routed?

So far, we have identified edges of information flow, finding that the aggregation step suggested by prior work is causally important to task success, but that it is preceded by a contextualizing subcircuit. We next investigate which information is passed along the edges. We are interested not in all information, but only in information about the input that is *causally implicated* in the downstream prediction. We do this by performing targeted manipulations on the input to manipulate the information present in specific activations.

For each edge, we specifically contrast the following hypotheses: An edge from some few-shot example $(x_i, y_i)$ might transport causally relevant information about (1) the specific tokens, (2) the functional relationship between $x_i$ and $y_i$ (but not their token identities), (3) the type of the $x_i$ or $y_i$ tokens without regard to their functional relationship.

**Patching Methodology**  We vary the information about the input available to a particular edge by constructing a contrastive input sample in which certain aspects have been altered (Appendix H, Figure 22), and patching the K and V activations directly feeding into the edge with the corresponding activations computed on that contrastive input (Appendix, Figure 28). For instance, when patching an edge $y_i \rightarrow t_{N+1}$, we modify the key and value activations associated with the positions of $y_i$ across all attention heads in the model. These modifications occur specifically when the activations are queried from the positions corresponding to $t_{N+1}$. Patching is done in parallel in each layer and head. All patching is done within the overall circuit identified for the task, i.e., edges not present in the circuit are ablated, and paths not present in the overall circuit play no role even in the patched version. For instance, to determine whether the edges $y_i \rightarrow t_{N+1}$ transport contextualized information about preceding few-shots $(x_{i'}, y_{i'})$ $(i' < i)$, we patch all inputs to this edge at position $y_i$ with activations computed on an input where the preceding few-shots have been corrupted; by varying the degree of corruption, we analyze which specific information is transported (Appendix H, Figure 22). We note that this methodology allows us to focus on *causally relevant* information used by downstream components. It is thus fundamentally different from supervised probing, which does not distinguish between information used or unused by downstream components. We focus on the 2B model for tractability.

**AGGREGATION Subcircuit**  We first investigated the information transported by the edges in the AGGREGATION subcircuit, consisting of the $x_i \rightarrow y_i$, $x_{N+1} \rightarrow t_{N+1}$ and $y_i \rightarrow t_{N+1}$ edges. Prior work (Cho et al., 2025; Kharlapenko et al., 2025; Wang et al., 2023) suggests that the $y_i \rightarrow t_{N+1}$ edges aggregate task information from the examples. Indeed, when we patched these with arbitrary $x_i$ or $y_i$ (which disrupts the functional relationship), functionality was largely destroyed (see Figures 3 and 8 for all patching results). In contrast, when we patched the $y_i \rightarrow t_N$ edges with other few-shots that are valid for the task, functionality was preserved. We also verified that the $x_i \rightarrow y_i$ edges transport $x_i$: patching these edges with

corrupted $x_i$ leads to total failure. This is to be expected, as in the simple AGGREGATION subcircuit, these edges are the only way for $x_i$ to influence the output. These results confirm prior proposals about the aggregation step (Cho et al., 2025; Kharlapenko et al., 2025; Wang et al., 2023) with rigorous causal analysis.

Our methodology allowed us to obtain two further findings. First, the $y_i \to t_{N+1}$ edges causally transport only task information and no token information. Namely, even in the presence of contextualizing edges, i.e., even though there are multiple paths from few-shots to the $t_{N+1}$, performance did not deteriorate even when the $y_i \to t_{N+1}$ and contextualizing edges transported information from different prompts when they were valid for the task.[9]

Second, we next asked whether the edges rely on the functional relationship between $x_i$ and $y_i$ or only their semantic types. We patched with pairs where the semantic types were preserved but the functional relationship had been disrupted (e.g., in Country-Capital, "France\tLondon", "Canada\tBeijing", etc. Appendix H, Figure 22). In four tasks, this led to a failure, confirming that the functional relationship is transported by $y_i \to t_{N+1}$. The one exception is the Person-Sport task: here, functionality is largely preserved ($> 90\,\%$ of the full accuracy) as long as the patched $x \in \mathcal{X}$, $y \in \mathcal{Y}$. This phenomenon provides an understanding of why ICL has sometimes been observed to be robust to perturbed labels (Min et al., 2022): for some tasks (in this case, the Person-Sport task), the task information is inferred on the basis of only the semantic types, without regard to their functional relationship.[10] In the context of the overall circuit:

> The $y_i \to t_{N+1}$ edges generally move information about the function $f$ from individual examples $(x_i, y_i)$ to $t_{N+1}$. In the Person-Sport task, they move information about $\mathcal{X}$ and $\mathcal{Y}$ without regard to functional relationship.

In order to understand the role of these components further, we investigated the functional behavior of the model with ablations applied. Overall, we observed three *fallback tasks* which accounted for most interpretable errors: *copying the query*, *producing from the correct output space*, *copying another token from the prompt*. With full ablation of the $y_i \to t_N$ edges, behavior was often accounted for by copying the query or copying another token from the corrupted input prompt (Appendix H, Figures 14, 15, 16, 17, 18). We note that as the corrupted prompts maintain the separators, the last separator position can still infer the presence of repeated structure, potentially explaining the presence of copying behavior. When patching the edge while preserving the output space, behavior often amounted to reproduction from the correct output space when there is well-delineated output space (e.g., Capitalization, Country-Capital, Person-Sport), or copying from the corrupted prompt when there is none (Copying). This is expected, as such prompts maintain repeated structure in terms of a specific output space, though with disrupted functional relationship.

**Is the ICL aggregation circuit a type of induction circuit?** An interesting question is whether attention in the $y_i \to t_{N+1}$ edges depends on the specific inputs, or just on the few-shot template. Induction circuits – argued to be essential for ICL by Olsson et al. (2022); Crosbie and Shutova (2024) – involve attention specifically to preceding position preceded by the same as the current token, one might thus hypothesize that heads involved in aggregation specifically attend to few-shot examples similar to the query. Indeed, Cho et al. (2025) argued that, at least in the text classification tasks they studied, the ICL circuit is a type of induction circuit, and the strongest information flow came from few-shot examples where the input texts are semantically closest to the query text. However, this argument was based on representational similarity, not any direct causal evidence. Relatedly, theoretical constructions based on linear transformers performing ICL for linear functions (e.g. Von Oswald et al., 2023; Vladymyrov et al., 2024; Mahankali et al., 2023) also crucially

---

[9]We emphasize that this is a claim about information that has a causal downstream effect in the circuit, not about information that can be obtained via probing, which is likely to be richer and include token information.

[10]An indirect argument based on representational similarity is made by Cho et al. (2025); our analysis improves by causally localizing the invariance to perturbed input-output mappings to the $y_i \to t_{N+1}$ edges, ruling out a role for other pathways.

assume that attention is strongest to examples where $x_i$ is similar to the query $x_{N+1}$, in line with the induction circuit. We thus investigated patching the key or query vectors involved in these attention edges with other valid prompts (preserving functional relationship but changing the specific tokens). Strikingly, across all five tasks, at both 3 or 10 examples, there was almost no discernible effect on accuracy (Table 4). This shows that, at least in the tasks and model considered here, the aggregating attention edges do not exhibit the functionality expected of the classical induction circuit: Attention is *not* preferentially given to specific few-shots on the basis of similarity to the query.[11]

**CONTEXTUALIZATION Subcircuit**   We next moved to analyzing the information transported by the contextualizing edges, starting with the edges connecting $y$'s ($y_i \to y_{i+1}$ or general $y_i \to y_j$). We found that these edges transport both information about $f$ and about $\mathcal{Y}$. Patching the $y_i \to y_{i+1}$ edges for Country-Capital only decreases accuracy significantly when $y$ leaves $\mathcal{Y}$, but not under other corruptions (Figure 3). In Capitalization, at 3 shots, replacing with $y \in \mathcal{Y}$ hurts much less than a general ablation of $y$ (accuracy 81% vs 56 %; Appendix F, Figure 8), which is as harmful as full corruption of these edges and leads to a large increase in copying behavior; nonetheless, accuracy still drops compared to ablations keeping $(x, y) \in f$. At 10 shots, Capitalization only responds to ablations where $y$ leaves $\mathcal{Y}$. In both cases, ablating the output space information leads to an increased number of copy-type mistakes (Appendix H, Figure 19). As noted before, edges do not provide specific token information used by downstream components at $t_N$: we verified that a token mismatch created by intervening on $(x_{i-1}, y_{i-1})$ for all $y_i \to y_j$ edges but not the $y_i \to t_{N+1}$ edges has no discernible effect on accuracy if the functional relation between is preserved by the prompt used for intervention. Overall, our causal interventions warrant the conclusion:

> The $y_i \to y_j$ / $y_i \to y_{i+1}$ edges contextualize the representation of each few-shot example $y_i$ by transporting the type of $\mathcal{Y}$, and sometimes the task $f$, from $(x_j, y_j)$ ($j < i$).

We next studied the edges connecting inputs $x$'s. Ablating these on Present-Past and Person-Sport turns out to not lead to a statistically significant drop, but the drop is substantive in Country-Capital (Figure 3). Here, ablations only hurt when $x$ left $\mathcal{X}$, showing that $x_i \to x_j$ transports the input space, but not (causally) the token $x_i$. Such ablations again lead to a substantial fraction of copying responses, i.e., the functional input-output behavior is impacted (Appendix H, Figure 20). Patching prior $x_i$'s for these edges, we found that they affect the final prediction both via $x_i \to x_{i+1} \to y_{i+1} \to t_{N+1}$ and via $x_N \to x_{N+1} \to t_{N+1}$ edges (Appendix, Table 8). Overall:

> The $x_i \to x_{i+1}$ edges contextualize the representations of $x_{N+1}$ and of individual few-shot outputs $y_i$ with information about $\mathcal{X}$.

### 3.3   A CONTEXTUALIZATION subcircuit is important in the presence of ambiguity

We conducted a controlled experiment aiming to stress-test the aggregation step. We designed the following hard version of the Present-Past task. We take advantage of the fact that some English verbs have identical present and past tense forms (e.g., "put", "spread"). In a prompt for the Present-Past task, examples using such verbs are thus ambiguous between the Present-Past task and a simpler Copying task (Appendix H, Figure 23.1). In this experiment, most individual examples do not give full task information, and may in fact provide misleading information. We hypothesize that direct aggregation of uncontextualized per few-shot representations as hypothesized by prior work may be less useful in this case, because

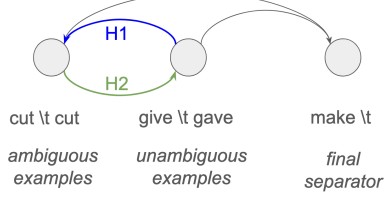

Figure 4: Two hypotheses for the causal graph on ambiguous inputs.

---

[11]Nonetheless, $t_{N+1}$, collecting information from few-shot outputs that followed the same separator token \t, might still rely on induction-head-like behavior, with the single preceding token as the key.

individual examples might, on their own, provide conflicting task information. The contextualization step might help resolve this ambiguity before the aggregation step. Even when 7 out of 10 few-shot examples are ambiguous, the full model continues to perform the task at very high accuracy (95%). Whereas the aggregation (AGGREGATION)-only circuit had performed at 93% on the standard Present-Past task (and similarly on the Copying task), its performance on the ambiguous prompts dropped to 56% (Appendix H, Table 9).

Wrong outputs largely consist of copying, expected as the ambiguous examples are compatible with the Copying task. Note that, in the AGGREGATION-only circuit, the activations at $y_i$ feeding into $y_i \rightarrow t_N$ are a mixture of activations for Copying or Present-Past; they each largely succeed in inducing the correct prediction on their own without contextualization when the task is consistent across the prompt, but $t_{N+1}$ fails at resolving conflicting task information when the task information is mixed across the prompt.

We found that contextualization is needed to explain the model's accuracy. Adding $x_i \rightarrow x_{i+1}$, $y_i \rightarrow y_{i+1}$ edges recovered part of the drop, but still underperformed the full model (68%). We next investigated which further edges help close the gap to the full model; indeed, in contrast to the standard tasks, we found separators now to play a role, as the full model with only separator-involving edges ablated performs at 85% (Appendix H, Table 9). Overall, adding a contextualization subcircuit with edges $y_i \rightarrow y_j$, $y_i \rightarrow t_j$, $x_i \rightarrow t_j$, $t_i \rightarrow y_i$, $t_i \rightarrow t_{i+1}$, $x_i \rightarrow x_{i+1}$ recovered almost all of the full model's performance (89.9%).

We investigated two hypotheses as to how contextualization resolves ambiguity (Figure 4):

1. (H1) Ambiguous examples obtain information from unambiguous examples, contextualizing their own representations so that they encode the Present-Past task instead of the Copying task.

2. (H2) Unambiguous examples obtain information from ambiguous examples, contextualizing their own representations so they can override misleading functional information from ambiguous examples.

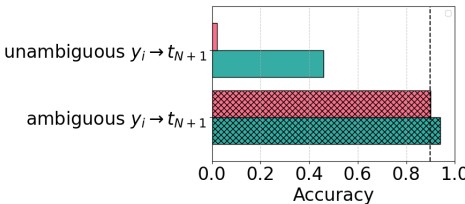

Figure 5: Patching $y_i \rightarrow t_{N+1}$ connections on only ambiguous examples (bottom) using activations taken on all-ambiguous (red) or all-unambiguous (green) barely hurts. In contrast, performing these patches on only unambiguous examples (top) leads to a drastic drop, even when patching with an all-unambiguous prompt (green).

We note that these hypotheses are not exclusive and might be true at the same time. To distinguish between them, we patched the K and V activations at either all ambiguous or all unambiguous examples feeding into $y_i \rightarrow t_{N+1}$ edges, with activations computed from either all-ambiguous or all-unambiguous prompts, in both cases erasing the presence of the other type of example (see Appendix H, Figure 23.2). Patching the $y_i$'s in ambiguous examples had little effect on the accuracy (Table 3, Figure 5). This suggests that, even if these examples may receive information about the presence of Present-Past examples, this is not causally important to task disambiguation.

In contrast, patching the unambiguous examples had great effect: Accuracy dropped to almost zero when patching with ambiguous prompts (in which case all information would be routed through the ambiguous examples, but we saw that they do not causally provide disambiguation signal via $y_i \rightarrow t_{N+1}$). Strikingly, accuracy dropped drastically even when patching with fully unambiguous prompts, showing that contextualizing the unambiguous examples with awareness of the presence of ambiguous examples is key to downstream performance. Overall, our results allow us to reject H1. They are as expected under H2:

In the presence of ambiguous examples, contextualization changes the representations of unambiguous few-shot examples. These altered representations of unambiguous examples are key to transporting task information to $t_{N+1}$, overriding potentially misleading information from ambiguous examples.

## 4 Discussion

On five tasks, we have shown that Gemma-2 2B performs ICL using a cascade contextualizing the representations of individual examples, and aggregating task information from them. Using rigorous causal interventions, we extend prior work on mechanistically understanding the mechanisms of ICL in transformers (e.g. Wang et al., 2023; Cho et al., 2025; Kharlapenko et al., 2025; Hendel et al., 2023; Todd et al., 2024; Bansal et al., 2022; Sia et al., 2024; Song et al., 2025; Yin and Steinhardt, 2025; Crosbie and Shutova, 2024). We extend on work on function or task vectors by showing that they are largely part of the aggregation circuit hypothesized by Cho et al. (2025); Kharlapenko et al. (2025). Most importantly, we establish the importance of a contextualization step, especially in the presence of ambiguous examples in a prompt.

Our work is distinct from Cho et al. (2025) in providing a computation graph supported by causal effects rather than representation similarity. A major difference between causal experiments in our work and Cho et al. (2025) is that we construct a circuit sufficient for recovering model performance, whereas Cho et al. (2025) zeroed out edges to confirm their importance. A major difference to Kharlapenko et al. (2025) is that we keep different few-shot examples distinct in our circuits, allowing us to identify the contextualization step.

Theoretical work has considered ICL both from the perspective of learning simple parametric functions (e.g. Akyürek et al., 2023; Von Oswald et al., 2023; Mahankali et al., 2023), and as general Bayesian inference (e.g. Xie et al., 2022; Wies et al., 2023; Hahn and Goyal, 2023). Theoretical constructions for transformers performing ICL often correspond to the AGGREGATION subcircuit (Akyürek et al., 2023; Mahankali et al., 2023), but contextualization of the few-shot inputs (Von Oswald et al., 2023; Vladymyrov et al., 2024) or outputs (Chen et al., 2024) does play a role in some constructions (see Appendix A for more), in agreement with our findings in an LLM. Mechanistic understanding of ICL also has the potential to explain some of its behavioral aspects. For instance, we found that, in the Person-Sport task, the circuit only assembles input and output spaces without regard to functional relationships, explaining the robustness to label noise sometimes observed (Min et al., 2022).

Our work is grounded in a tradition of identifying important components in neural networks (e.g. Michel et al., 2019). Modern circuit discovery in LMs was pioneered by Wang et al. (2022); Hanna et al. (2023); a unifying framework was provided by Conmy et al. (2023). Scaling this to larger models (e.g. Syed et al., 2023; Hanna et al., 2024; Ferrando and Voita, 2024) is a focus of recent research.

A limitation of the present study is that, to keep computational cost manageable, it focuses on one model family and five tasks. A second limitation is that it focuses on causal understanding of information flow between positions, and does not interpret MLPs or individual heads. Future work might combine our results with a further scaled-up version of the sparse feature circuits of Kharlapenko et al. (2025) to understand contextualization at the level of individual representations. It is likely that further advances in interpretability are needed to comprehensively interpret all individual components in models of the scale studied here.

## 5 Conclusion

We have conducted extensive causal interventions to understand the flow of information from individual few-shot examples to the next-token prediction in ICL, at the example of five tasks in Gemma-2 2B. Our most important result is to establish a two-stage procedure, whereby the model contextualizes representations of individual few-shot examples, which are then aggregated to prepare next-token prediction.

## Author Contributions

AB led the project and conducted the experiments. YV contributed to scaling experiments. XH contributed to discussions throughout the project. MH supervised the project, provided input on the experimental design, and drafted the paper text. All authors refined the paper.

## Acknowledgments

Funded by the Deutsche Forschungsgemeinschaft (DFG, German Research Foundation) – Project-ID 232722074 – SFB 1102. We thank Entang Wang for comments on the manuscript.

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

## A   ICL on Linear Regression Benefits from Contextualization Step

A substantial amount of work has studied ICL in small transformers trained to solve simple synthetic tasks, particularly linear regression. However, the link between these mechanisms and the ICL circuits inside LLMs largely remain unknown. Theoretical constructions for ICL on linear regression based on gradient descent (GD) and similar algorithms often involve just multiple rounds of parallel aggregation from each few-shot; in particular, a single round of GD can be implemented as a single aggregation step, and is the optimal ICL strategy for a one-layer (linear) transformer (Mahankali et al., 2023). However, there is theoretical reason to believe that a contextualization step is beneficial in preprocessing the representations of individual few-shots before aggregation. First, multilayer linear transformers implement the GD++ algorithm (Von Oswald et al., 2023), which iteratively contextualizes $x_i$'s to approximate the input data covariance (Von Oswald et al., 2023), and which allows faster convergence than GD (Vladymyrov et al., 2024); such preconditioning also appears in constructions for ICL on non-isotropic data (Ahn et al., 2023). Second, Chen et al. (2024) showed that preprocessing of individual few-shots is beneficial in *sparse* linear regression. While these works show that contextualization helps outperform gradient descent, they do not establish that the step is needed for optimal performance, as *a priori* there might be undiscovered algorithms implementable by transformers outperforming GD with only the aggregation step.

We empirically investigated which attention edges are needed when transformers learn linear regression in context. Following Garg et al. (2022), we trained transformers to perform linear regression, i.e., each prompt has a latent task $v \in \mathbb{R}^d$ and each few-shot consists of a vector $x \in \mathbb{R}^d$ and the output $y = v^T x \in \mathbb{R}$. We chose $d = 20$.

We investigated three subcircuits:

1. AGGREGATION: $x_i \rightarrow y_i, y_i \rightarrow x_{N+1}$
2. Between-Xs: $x_i \rightarrow x_j$ (all $1 \leq i < j \leq N + 1$)
3. Between-Ys: $y_i \rightarrow y_j$ (all $1 \leq i < j \leq N$)

We trained the model with the aggregation-only circuit, and three combinations of this and the other two subcircuits (Figure 6). As we trained from scratch, we ablated edges simply by zeroing out their attention weights throughout training.

Garg et al. (2022) trained the model on the aggregation of the losses of each $y_i$ ($i \leq 40$) conditioned on the prior examples. This was not feasible in our setup, as the set of ablated edges varies with the target. We instead randomized the length $N$ of the prompt in every training step (between 10 and 40) and trained only on the last output ($y_{N+1}$) on each prompt. As there are no separators, the position at which this output is predicted is the query, $x_{N+1}$.

We varied the depth (4, 6, 8, 12 layers). The model of Garg et al. (2022) had used 12 layers; we reasoned that contextualization might be more important in shallow models because these provide less opportunity for deep computations at the last position We used the original

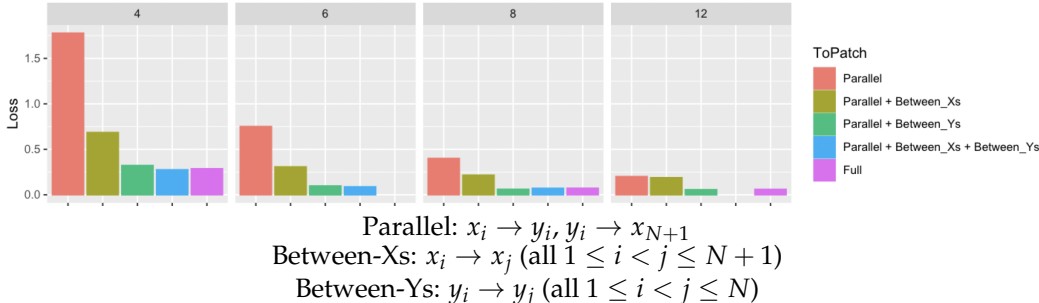

$$\text{Parallel: } x_i \rightarrow y_i, \ y_i \rightarrow x_{N+1}$$
$$\text{Between-Xs: } x_i \rightarrow x_j \ (\text{all } 1 \le i < j \le N+1)$$
$$\text{Between-Ys: } y_i \rightarrow y_j \ (\text{all } 1 \le i < j \le N)$$

Figure 6: Loss (MSE) on ICL on linear functions when trained with attention links ablated during training, with transformers with $l = 4, 6, 8, 12$ layers. The AGGREGATION circuit performs much better than a random baseline (loss 20.0), but underperforms the full model by a large margin in shallow transformers. Adding links within $x$'s or $y$'s improves performance, particularly the latter. Adding both performs at par with the full model, in line with our findings on real-world tasks in Gemma-2.

code from Garg et al. (2022), but increased the batch size to 1024 and decreased the number of steps to 150K for tractability. The model was run on one NVIDIA H100 card.

Results are in Figure 6. The AGGREGATION subcircuit performs much better than random baseline (MSE loss of 20), but suffers substantially for shallow models. $x_i \rightarrow x_j$ edges improve, but cannot close the gap to the full model. $y_i \rightarrow y_j$ edges nearly or fully close the gap. Overall, while parallel aggregation plays an important role, a contextualization step is needed to close the gap to the full transformer.

## B  Computational Resources

For the 2B model, a single patching run takes roughly 40 minutes on 1 NVIDIA A100 40GB GPU on one 3-shot task, and roughly 4 hours on a 10-shot task. Costs are higher on the 9B and 27B models.

## C  Datasets

We took Person-Sport, Country-Capital and Present-Past datasets from Todd et al. (2024), and selected words for Capitalization dataset randomly from 5000 English words. For the Copying dataset we used the same source as for Capitalization.

All datasets were randomly divided into train and test parts. To construct an ICL prompt, we select examples from the train split of the dataset without replacement to use as few-shot examples, and one example from the test split to use as final query and target. We used a test size of 100 examples in each dataset.[12] Train sizes of datasets range from 39 to 4000.

For the ambiguity dataset, we selected 20 English verbs with identical present and past tense forms, and combined them with the Present-Past dataset. For each prompt, we randomly selected $N = 7$ positions in the Present-Past prompt to replace with ambiguous examples.

## D  Patching methodology

We use path patching (Wang et al., 2022; Hanna et al., 2023) to identify position-level circuits and analyze information flow along edges. This method involves: (1) representing the model as a computation graph with token embeddings as inputs, loss value as output, and

---

[12]The high computational cost of the full set of patching experiments (Figure 8) prevented us from running larger test sets. Importantly, our key conclusions are drawn only from patching effects with statistically significant changes in accuracy (Figure. 8).

specific activations as nodes; (2) ablating graph components to isolate those responsible for specific behaviors. This section details our computation graph definition (nodes, edges), loss function and ablation implementation.

**Computation graph**  Nodes consist of activations at the input/output of each attention head across positions. Each node is represented as a tuple (*input/output*, layer, head, position), where "position" indicates a token's semantic role (e.g., $x_1$, $y_3$, $t_2$, $n_4$). Note that $x_i$ and $y_i$ positions typically span multiple tokens but are treated as single nodes. Each node is thus a tensor of size (number of tokens in position, hidden dimension).

Edges represent computational connections between nodes. In our position-level circuits, edges exist exclusively within attention heads, as only these transfer information between positions (Appendix E details edge inclusion criteria). Removing an edge between (*input*, L, H, pos=$x_1$) and (*output*, L, H, pos=$y_1$) disconnects these nodes, implemented by computing:

(*output*,L,H,pos=$y_1$)=Attention((*input*,L,H,pos=$x_1$)$_C$, (*input*,L,H,pos=$t_1$), (*input*,L,H,pos=$y_1$))

instead of

(*output*,L,H,pos=$y_1$)=Attention((*input*,L,H,pos=$x_1$), (*input*,L,H,pos=$t_1$), (*input*,L,H,pos=$y_1$))

where subscript $C$ denotes counterfactual activations.

When discovering position-level circuits, we abstract all edges between the same pair of positions into a single edge. Specifically, we treat all edges of the form ((*input*, $L$, $H$, pos=$A$) → (*output*, $L$, $H$, pos=$B$) for all $L$, $H$) as a single edge: (*input*, pos=$A$) → (*output*, pos=$B$). Thus, removing such an edge corresponds to removing all edges between positions $A$ and $B$ across all attention heads and layers.

We use cross-entropy loss over all tokens in the output position ($y_{N+1}$). Performance drop is measured as accuracy decrease between ablated and normal model runs.

**Ablation Procedure**  Edge ablation requires two forward passes (Fig. 28):

1. Run counterfactual input $I_C$, saving all input node activations (*input*, L, H, pos)$_C$
2. Run correct input $I$. For each attention head, compute outputs position-wise, which requires as many passed through attention block as there are positions. When ablating edge $A \rightarrow B$ in head $H$ (layer $L$), for position $B$: substitute activation at $A$ with (*input*, L, H, A)$_C$ saved from Step 1. Concatenate position outputs into final tensor.

With this methodology we can identify *circuits* - subgraphs of the model's computation graph that can perform the task with high accuracy when the edges outside the circuit are ablated.

Our approach to circuit discovery is closely related to the recent work of Haklay et al. (2025), with several key differences:

- The dataset schema in our case is defined by the specifics of the task: what we refer to as x's, y's, t's, and n's corresponds to what Haklay et al. (2025) call a "span."
- We define an edge from position $A$ to position $B$ as patching both the V and K activations at position $A$ when queried by position $B$, whereas Haklay et al. (2025) distinguish between edges originating from Q, K, and V activations.
- We do not take edges from a position to itself into account and thus do not have separate nodes for MLPs, embeddings and logits.
- To identify position-level circuits, we abstract all edges from position $A$ to position $B$ across all layers into a single edge. We only distinguish between layers when identifying activation-level circuits (Section 3.1).

For analyzing information routing through specific edge sets (Section 3.2), we partition edges into three categories: (1) Outside circuit, (2) Inside circuit but not under investigation, (3) Inside circuit and under investigation. The extended procedure adds a third step:

1. Run counterfactual input $I_C$, saving $(input, L, H, pos)_C$
2. Run semi-counterfactual input $I_{SC}$, ablating edges outside circuit (Set 1). Save input node activations for heads corresponding to Set 3 edges: $(input, L, H, pos)_{SC}$
3. Run correct input $I$ while ablating:
   - Set 1 edges using $(input, L, H, pos)_C$
   - Set 3 edges using $(input, L, H, pos)_{SC}$

This isolates performance changes caused by removing only Set 3 information not present in the semi-counterfactual. Counterfactual inputs (e.g., replacing "Berlin" with "table") remove more information than semi-counterfactual (e.g., replacing "Berlin" with "Paris"), which preserve broader categorical attributes. This reveals which information aspects along specific edges impact downstream predictions.

**Handling Multiple Tokens per Position** Our approach requires identical token counts per position in counterfactual and correct prompts, enforced during construction. When ablating edges from position $A$, we replace its entire activation tensor with the counterfactual version.

For projecting position-level circuits to activation-level circuits, we assign each edge a single importance score by summing scores across all constituent token-level edges. Position-level circuits require no such aggregation. See Appendix F for activation-level circuit details.

# E  Overall Set of Edges

We focus our discussion on edges involving $x_1, y_1, \ldots, x_{N+1}, t_{N+1}$, specifically on edges going from one of these positions to a different one. Here, we exactly document which edges remain un-ablated in our experiments with position-level circuits:

1. Edges involving bos/eos/pad tokens are always ablated.
2. We keep edges $x_{N+1} \rightarrow y_{N+1}$ and $t_{N+1} \rightarrow y_{N+1}$, and do not ablate these. This is because these edges support autoregressive prediction of the response $y_{N+1}$ when it consists of more than one token.
3. We always include edges going from one position to itself, provided the position plays a role in the circuit at all. Note that a single position may be occupied by multiple tokens (that is, if $x_i$ or $y_i$ has more than one token); in this case, such edges can go from one token to another, without leaving a position in the position-level circuit.
4. We consider only edges between inputs and outputs of attention heads, as we focus on information flow between positions.

Note that our visualizations of circuits only include positions from $x_1$ to $t_{N+1}$, as positions before or after are not connected to the $N$ few-shot examples by any edges; recall that our focus is on the process by which information is assembled from the few-shot examples.

# F  Details for Gradient-Based Circuit Finding Algorithm

Here, we describe how we identified activation-level circuits.

The algorithm we use consists of two parts:

1. (Head Pruning) Starting from the position-level circuits, we view the model as a computational graph where nodes are the outputs of attention heads indexed by $\langle position, layer, head \rangle$, and prune heads.
2. (Edge Pruning) We then refine this circuit by pruning edges. We view the model we get from the first step as a computational graph where edges connect inputs of attention heads to outputs of attention heads.

Note that we could have also applied edge pruning to obtain an activation-level circuit from the start, rather than going through a position-level and head-level circuit first. However, such a strategy might have resulted in a circuit with less systematic patterns in which pairs of positions have edges. In contrast, going via position-level circuits allows us to jointly interpret all edges linking two positions, making interpretation much more feasible given the large number of heads and edges involved in the circuit.

### F.1 Step 1: Head pruning

**Definition of Computation Graph** We first view the model as a computation graph where nodes are indexed by $\langle \text{position}, \text{layer}, \text{head} \rangle$ (position $\in \{x_1, y_1, \ldots, x_N, y_N, x_{N+1}, t_{N+1}, y_{N+1}\}$, layer $\in \{1, \ldots, n_{layers}\}$, head $\in \{1, \ldots, n_{heads}\}$). Notably, our circuits also include $y_{N+1}$ even though in principle the last separator is where the answer is predicted, which is in order to account for multi-token outputs. The loss for gradient-based pruning is the entire multi-token CE on the answer.

These nodes represent activations on the output of attention head after projection onto residual stream, but before post-attention layernorm.

**Pruning Strategy** The algorithm we use for head pruning is based on the *Head Importance Scores* method (Michel et al., 2019). We compute importance scores for each component in a computational graph, and then prune the components with smallest importance scores. Whereas the original Head Importance Score defined by Michel et al. (2019) approximates how much the loss would change if we change the output of one head to zero, we instead set the output of a head to the result of ablating it, that is, the value that it would have on a corrupted prompt. Thus, we adapt the importance score to approximate how much the loss would change if we change the output of the head from its clean activation $\xi_{clean}$ to the activation $\xi_{corr}$ computed on a corrupted prompt $prompt_{corr}$. The importance score of an attention head $h$ is then the following:

$$\mathcal{I}_h = \mathbb{E}_{prompt \sim X} \left| (\text{Att}_h(\xi_{clean}) - \text{Att}_h(\xi_{corr}))^T \frac{\partial \mathcal{L}(\xi_{clean})}{\partial \text{Att}_h(\xi_{clean})} \right|$$

where $\mathcal{L}(\xi_{N+1})$ denotes the cross-entropy loss on predicting the correct response given a prompt. We arrive at this by performing a Taylor approximation of the loss after corrupting the activation:

$$\mathcal{L}(\xi_{corr}) \approx \mathcal{L}(\xi_{clean}) + (\text{Att}_h(\xi_{corr}) - \text{Att}_h(\xi_{clean}))^T \frac{\partial \mathcal{L}(\xi_{clean})}{\partial \text{Att}_h(\xi_{clean})}$$

and rearranging to obtain:

$$|\mathcal{L}(\xi_{corr}) - \mathcal{L}(\xi_{clean})| \approx \left| (\text{Att}_h(\xi_{clean}) - \text{Att}_h(\xi_{corr}))^T \frac{\partial \mathcal{L}(\xi_{clean})}{\partial \text{Att}_h(\xi_{clean})} \right|$$

When a head belongs to multiple tokens within a position (a multi-token $x_i$ or $y_i$), importance scores are summed.

We then score each head with this importance score and "ablate" 20% of the remaining heads that have the lowest score. "Ablating" in our setting is setting the output of a head to the one it would have on $x_{corr}$. Then we calculate the scores for the model without pruned heads again and prune another 20% of the heads using the newly calculated scores.

We stop when either the loss increases more than twice or the accuracy drops by more than 10%, compared to the original position-level circuit.

We take the last checkpoint for which the loss and accuracy were still inside the threshold.

### F.2 Step 2: Edge pruning

In the second step, we view the model we get from step (1) as a computational graph where edges connect inputs of attention heads before layernorm and outputs of attention heads before layernorm, but after projection to residual stream.

| | Cap | | CC | | PP | | Copy | | PS | |
|---|---|---|---|---|---|---|---|---|---|---|
| n-shot | 3 | 10 | 3 | 10 | 3 | 10 | 3 | 10 | 3 | 10 |
| (1) add $xs \rightarrow xs$ | 0.54 | 0.53 | 0.77 | 0.81 | 0.87 | 0.95 | 0.87 | 0.95 | 0.84 | 0.88 |
| (2) add $ys \rightarrow ys$ | 0.89 | 0.97 | 0.44 | 0.61 | 0.90 | 0.94 | 0.94 | 1.00 | 0.80 | 0.82 |
| (3) add $xs \rightarrow ys$ | 0.50 | 0.55 | 0.36 | 0.29 | 0.83 | 0.91 | 0.88 | 0.96 | 0.82 | 0.82 |
| (4) add $ys \rightarrow xs$ | 0.51 | 0.51 | 0.70 | 0.82 | 0.88 | 0.94 | 0.89 | 0.94 | 0.84 | 0.83 |
| AGGREGATION | 0.53 | 0.55 | 0.40 | 0.41 | 0.84 | 0.93 | 0.91 | 0.96 | 0.83 | 0.80 |

Tasks: Cap = Capitalization, CC = Country-Capital, PP = Present-Past, PS = Person-Sport

Table 1: Accuracy of Gemma2-2B on circuits when adding edges between different xs and ys combinations to the AGGREGATION-only circuit. Our aim is to find a minimal circuit sufficient for recovering most ($\approx 90\%$) of the full model's accuracy. Adding (2) most consistently improves accuracy compared to the AGGREGATION-only circuit, with a large increase on Capitalization, where the other types of edges achieve no such increase. Adding (1) or (4) both substantially improve accuracy over the AGGREGATION-only circuit in Country-Capital. We note that adding (1) results in a smaller number of possible paths than (4), which simplifies interpretation of information flow; hence, focused on (1). In other words, to keep the circuits tractable and understandable, we do not aim to guarantee completeness, but aim to provide minimal and sufficient circuits. Overall, we focused on (1) and (2) for determining position-level circuits. As we show, our resulting position-level circuits are indeed sufficient for recovering $\geq 90\%$ of the full model's accuracy (Figure 2).

Notice that all the edges except for edges inside attention are connecting only tokens of the same type. Since the information between tokens gets mixed in a model only inside attention layer, the edges between types are only edges between inputs and outputs of attention. Our interpretation focuses on information flow between different positions in the prompt, that is why we do not consider pruning other types of edges to get the activation-level circuits (Figures 9–13).

We use *edge attribution patching* (Syed et al., 2023) to prune edges. As in head pruning, we compute importance scores reflecting first-order approximations of the effect on the loss of ablating each edge and then "prune" the edges with the lowest importance. After "pruning" the edge, we replace it with the value on corrupted input (i.e., ablate it).

When we represent the model as computational graph, we view as nodes activations of inputs and outputs of attention heads divided by type of tokens.

We distinguish between 16 types of tokens: 3 types for queries in the few-shot examples, 3 types for targets in few-shot examples, 3 types for separators between queries and targets in few-shot examples, 3 types for separators between few-shot examples, a type for query of the whole input, a type for target that the model needs to generate, a type for the last separator before target, a type for bos/eos/pad tokens. In Figure 1 we use color coding to differentiate token types, though some are merged into single categories for visual simplicity. We view each activation site as 16 nodes that correspond to different token types.

**Stopping Criterion** We stop pruning edges based on a threshold on increased loss ($1.5 * loss\_before\_pruning$) and decreased accuracy ($0.9 * accuracy\_before\_pruning$). As above, thresholds were chosen in preliminary experiments and not extensively tuned; these could be refined in future work.

## G   Error Analysis

To obtain the model's (with or without ablations applied) output on a prompt, we greedily decoded next tokens after $t_{N+1}$ until a separator \n was generated. We classified the resulting answer for whether it

1. equals the ground-truth answer, or

| | | FULL | REMOVE-SEPS | AGGREGATE | AGGREGATE $+ y_i \to y_j$ | AGGREGATE $+ y_i \to y_{i+1}$ $+ x_i \to x_{i+1}$ |
|---|---|---|---|---|---|---|
| PP | 1-shot | 0.62 | 0.41 | 0.27 | 0.27 | 0.31 |
| | 3-shot | 0.96 | 0.92 | 0.84 | 0.90 | 0.93* |
| | 5-shot | 0.93 | 0.82 | 0.70 | 0.77 | 0.75 |
| | 7-shot | 0.96 | 0.93 | 0.88 | 0.89 | 0.89 |
| | 10-shot | 0.99 | 0.92 | 0.93 | 0.94 | 0.94* |
| CC | 1-shot | 0.80 | 0.62 | 0.23 | 0.23 | 0.46 |
| | 3-shot | 0.90 | 0.87 | 0.40 | 0.44 | 0.79* |
| | 5-shot | 0.85 | 0.82 | 0.36 | 0.26 | 0.66 |
| | 7-shot | 0.88 | 0.83 | 0.33 | 0.43 | 0.74 |
| | 10-shot | 0.88 | 0.85 | 0.41 | 0.61 | 0.80* |
| Copy | 1-shot | 0.99 | 0.87 | 0.91 | 0.91 | 0.85 |
| | 3-shot | 1.00 | 0.88 | 0.91* | 0.94 | 0.93 |
| | 5-shot | 0.93 | 0.92 | 0.92 | 0.90 | 0.93 |
| | 7-shot | 1.00 | 0.92 | 0.97 | 0.97 | 0.97 |
| | 10-shot | 1.00 | 0.92 | 0.96* | 1.00 | 0.96 |
| Cap | 1-shot | 0.39 | 0.29 | 0.54 | 0.54 | 0.53 |
| | 3-shot | 0.97 | 0.87 | 0.53 | 0.89* | 0.88 |
| | 5-shot | 0.80 | 0.70 | 0.31 | 0.74 | 0.66 |
| | 7-shot | 0.98 | 0.92 | 0.34 | 0.91 | 0.80 |
| | 10-shot | 0.99 | 0.91 | 0.55 | 0.97* | 0.94 |
| PS | 1-shot | 0.89 | 0.86 | 0.58 | 0.58 | 0.76 |
| | 3-shot | 0.86 | 0.88 | 0.83 | 0.80 | 0.83* |
| | 5-shot | 0.83 | 0.80 | 0.68 | 0.73 | 0.76 |
| | 7-shot | 0.87 | 0.89 | 0.78 | 0.77 | 0.81 |
| | 10-shot | 0.91 | 0.93 | 0.80 | 0.82 | 0.87* |

Tasks: Cap = Capitalization, CC = Country-Capital, PP = Present-Past, PS = Person-Sport

Table 2: Accuracy of circuits in Gemma2-2B, by prompt length.
FULL refers to the full model.
REMOVE-SEPS refers to the model with all edges involving separators other than $t_{N+1}$ ablated.
AGGREGATE refers to the circuit consisting of only $x_i \to y_i$, $y_i \to t_{N+1}$, $x_{N+1} \to t_{N+1}$ edges.
$y_i \to y_j$ refers to the edges $y_i \to y_j$ ($1 \leq i < j \leq N - 1$).
$y_i \to y_{i+1}$ refers to the edges $y_i \to y_{i+1}$ ($1 \leq i \leq N - 1$).
$x_i \to x_{i+1}$ refers to the edges $x_i \to x_{i+1}$ ($1 \leq i \leq N$).
For each task, we selected one of the three tasks. For each task, we use asterisks to mark the chosen position-level circuit. We chose the simplest circuit achieving $\geq 0.9$ at $N = 3$, or (if there is none) the circuit achieving highest accuracy at $N = 3, 10$. Circuits recover 94% Present-Past), 90% (Country-Capital), 96% (Copying), 97% (Capitalization) and 95% (Person-Sport) of the full model's accuracy. Compare Table 5 for the 9B model, and Table 6 for the 27B model. Compare Figure 7 for a visual representation.

| | | | |
|---|---|---|---|
| | | no corruption | 0.90 |
| $y_i \to t_{N+1}$ for all ambiguous $(x_i, y_i)$ | | fully ambiguous prompt | 0.90 |
| | | fully unambiguous prompt | 0.94 |
| $y_i \to t_{N+1}$ for all unambiguous $(x_i, y_i)$ | | fully ambiguous prompt | **0.02** |
| | | fully unambiguous prompt | **0.46** |

Table 3: Accuracy when patching the $y_i \to t_{N+1}$ edges separately for ambiguous and unambiguous examples. 10-shot setting, 7 ambiguous examples in each prompt. In bold - significant drop.

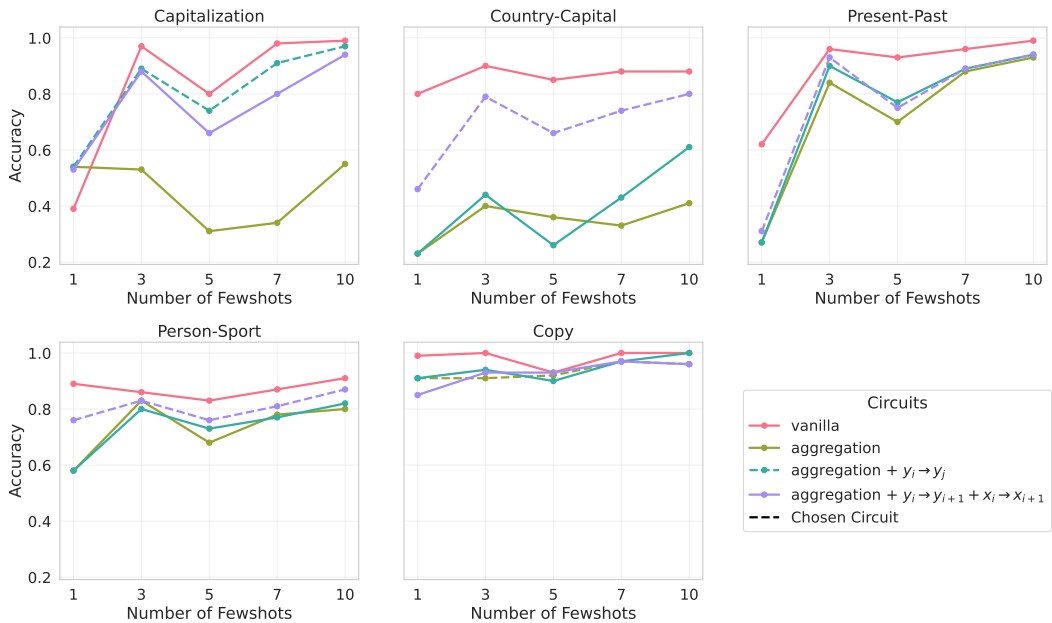

Figure 7: Accuracy of circuits by prompt length (Gemma-2 2B). Compare Table 2.

| $N$ | Present-Past 3 | Present-Past 10 | Country-Capital 3 | Country-Capital 10 | Copying 3 | Copying 10 | Capitalization 3 | Capitalization 10 | Person-Sport 3 | Person-Sport 10 |
|---|---|---|---|---|---|---|---|---|---|---|
| no ablation | 0.93 | 0.94 | 0.79 | 0.80 | 0.91 | 0.96 | 0.89 | 0.97 | 0.83 | 0.87 |
| key | 0.92 | 0.94 | 0.82 | 0.81 | 0.88 | 0.98 | 0.87 | 0.99 | 0.81 | 0.91 |
| query | 0.93 | 0.96 | 0.82 | 0.81 | 0.88 | 0.96 | 0.89 | 0.98 | 0.84 | 0.87 |

Table 4: Patching key and query vectors in $y_i \to t_{N+1}$ edges with other examples from the same task, at $N = 3$ and $N = 10$-shot prompts. Accuracy shows no clear change compared to the full circuit. This shows that these attention edges transport information about the prompt (and thus the task) only via the value vectors, but not the allocation of attention.

| n-shot | PP 3 | PP 10 | CC 3 | CC 10 | Copy 3 | Copy 10 | Cap 3 | Cap 10 | PS 3 | PS 10 |
|---|---|---|---|---|---|---|---|---|---|---|
| full model | 0.99 | 1.00 | 0.91 | 0.90 | 1.00 | 1.00 | 0.95 | 1.00 | 0.94 | 0.96 |
| REMOVE-SEPS | 0.97 | 0.98 | 0.90 | 0.90 | 1.00 | 1.00 | 0.91 | 0.97 | 0.92 | 0.96 |
| AGGREGATION | 0.79 | 0.84 | 0.73 | 0.78 | **0.99** | **1.00** | 0.27 | 0.16 | 0.83 | 0.85 |
| AGGREGATION $+ y_i \to y_j$ | 0.93 | 0.98 | 0.88 | 0.90 | 0.99 | 1.00 | **0.81** | **0.97** | 0.86 | 0.88 |
| AGGREGATION $+ y_i \to y_{i+1}$ $+ x_i \to x_{i+1}$ | **0.95** | 0.98 | **0.90** | **0.89** | 0.99 | 1.00 | 0.81 | 0.82 | **0.90** | **0.93** |

Tasks: Cap = Capitalization, CC = Country-Capital, PP = Present-Past, PS = Person-Sport

Table 5: Accuracy results of circuits on Gemma2-9B. Removing edges between the separators hurts even less than in the 2B model (compare Table 2 for that). As in the 2B model, accuracy of the AGGREGATION-only circuit is low on some tasks, in particular Capitalization. Including contextualization recovers almost the entire performance of the full model. In bold are the circuits we had selected based on the 2B model. Compare Table 2 for the 2B model, and Table 6 for the 27B model.

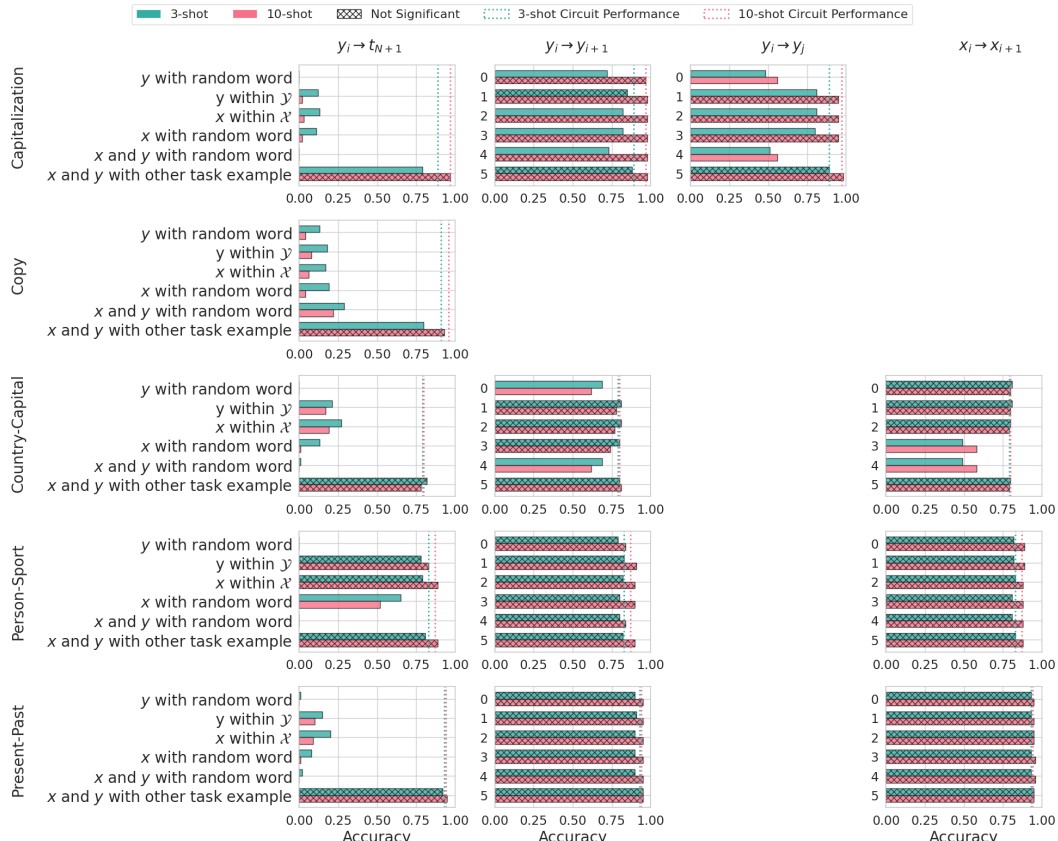

Figure 8: Accuracies in patching experiments within circuits to get information about edges semantics (Gemma-2 2B). For each ablation, we indicate whether the drop in performance is statistically significant according to a binomial test with $\alpha = 0.05$. We note that some components show no statistically significant drop when ablated; this is because we had conservatively selected maximally accurate circuits for each task. Note that $y_i \rightarrow y_{i+1}$ is a subset of $y_i \rightarrow y_j$; on Capitalization, the circuit includes the larger set.

| | PP | | CC | | Copy | | Cap | | PS | |
|---|---|---|---|---|---|---|---|---|---|---|
| n-shot | 3 | 10 | 3 | 10 | 3 | 10 | 3 | 10 | 3 | 10 |
| full model | 1.00 | 1.00 | 0.91 | 0.89 | 1.00 | 1.00 | 0.96 | 1.00 | 0.91 | 0.97 |
| REMOVE-SEPS | 0.99 | 0.99 | 0.90 | 0.90 | 0.97 | 1.00 | 0.92 | 0.95 | 0.89 | 0.96 |
| AGGREGATION | 0.87 | 0.88 | 0.84 | 0.80 | **0.97** | **0.99** | 0.38 | 0.34 | 0.84 | 0.86 |
| AGGREGATION $+ y_i \rightarrow y_j$ | 0.96 | 0.98 | 0.88 | 0.87 | 0.97 | 1.00 | **0.86** | **0.90** | 0.84 | 0.89 |
| AGGREGATION $+ y_i \rightarrow y_{i+1}$ $+ x_i \rightarrow x_{i+1}$ | **0.97** | **0.98** | **0.89** | **0.89** | 0.97 | 1.00 | 0.82 | 0.68 | **0.88** | **0.94** |

Tasks: Cap = Capitalization, CC = Country-Capital, PP = Present-Past, PS = Person-Sport

Table 6: Accuracy results of circuits on Gemma2-27B. In bold are the circuits we had selected based on the 2B model. Compare Table 2 for the 2B model, and Table 5 for the 9B model.

**Capitalization Task**

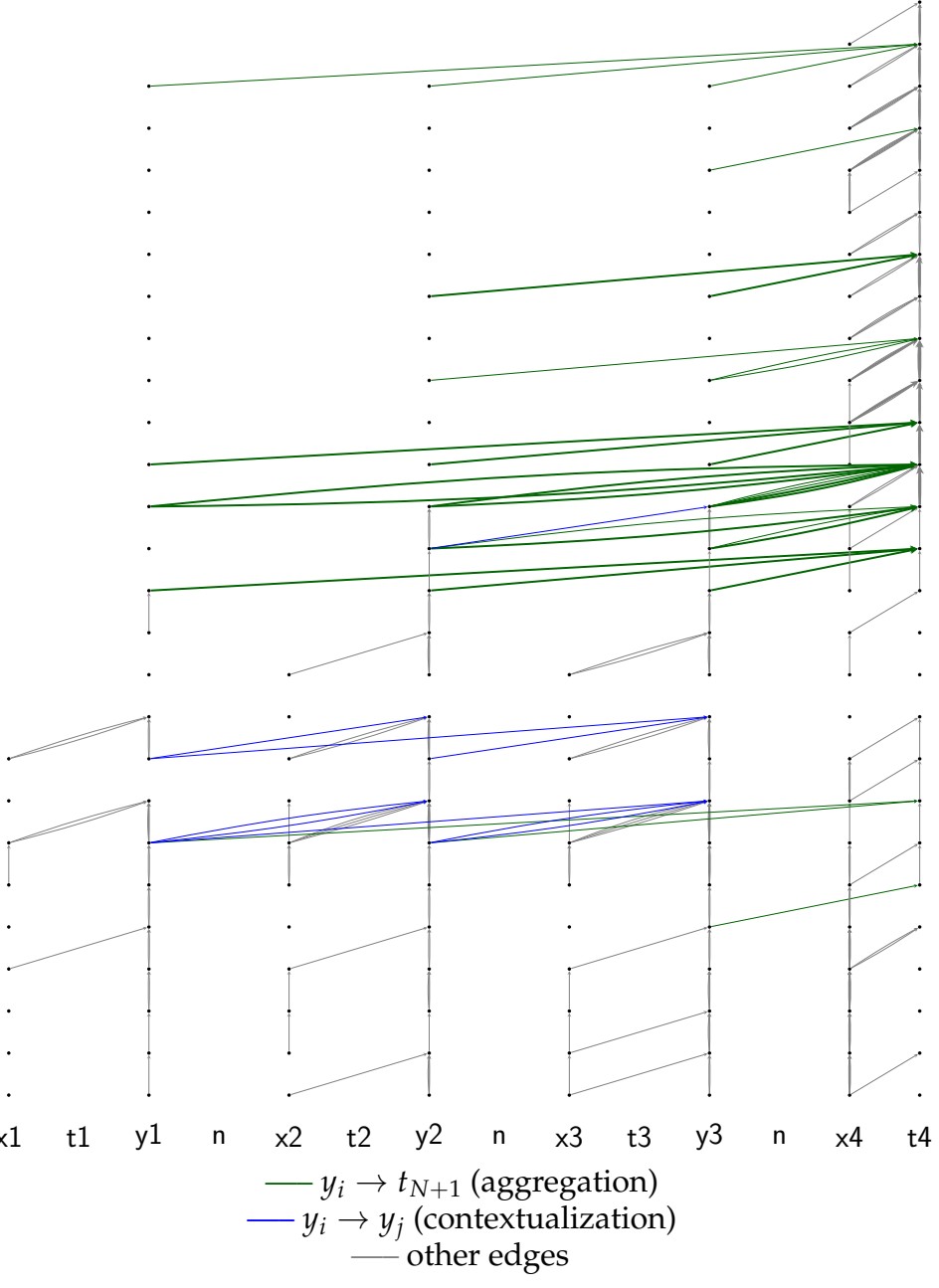

Figure 9: Activation-level circuit. Only attention edges are shown. Each row corresponds to one layer, starting from the input layer at the bottom. Edges to $t_{N+1}$ are marked in bold when they belong to an attention head with top-10 function vector score for the task.

**Country-Capital Task**

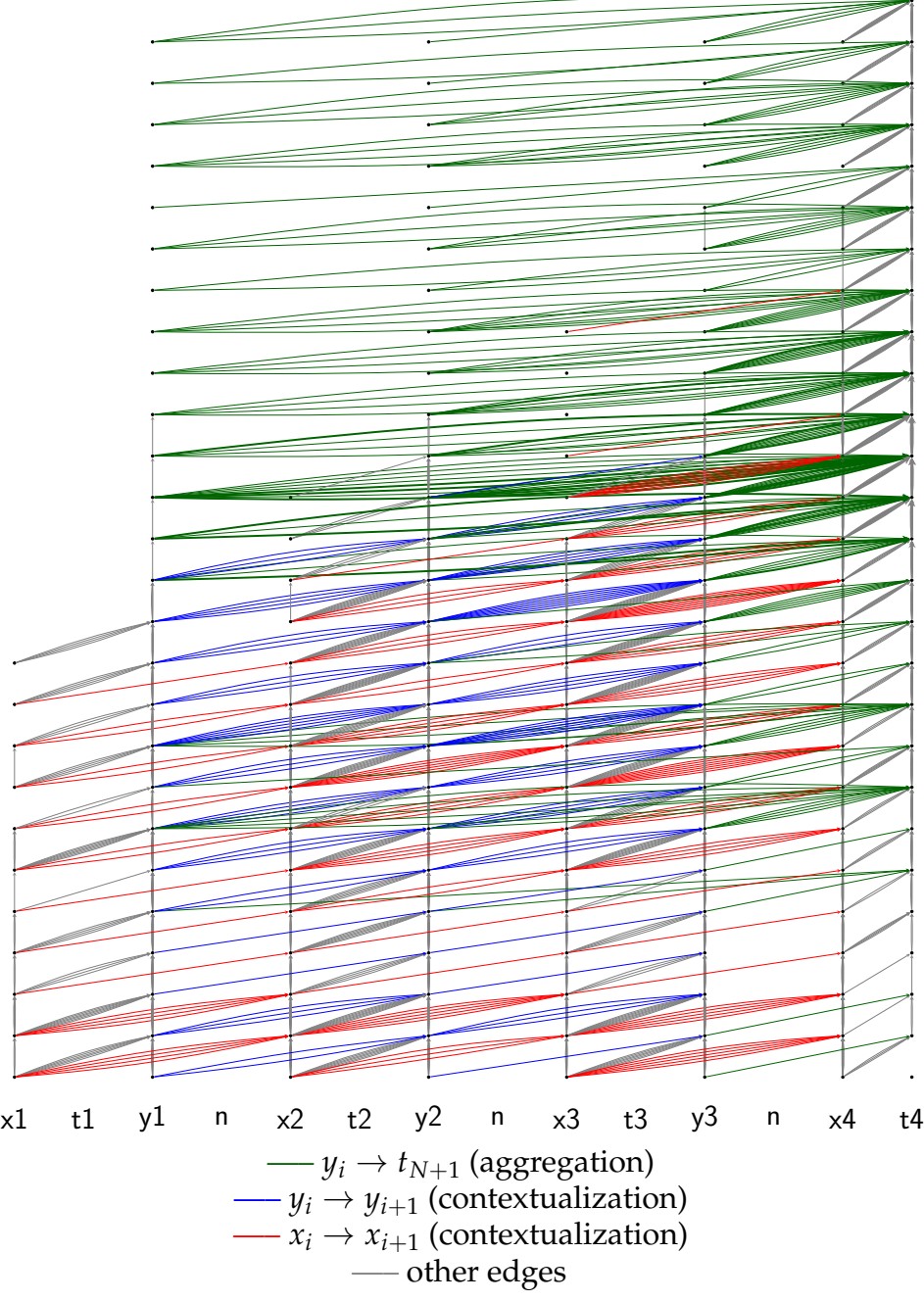

Figure 10: Activation-level circuit. Only attention edges are shown. Each row corresponds to one layer, starting from the input layer at the bottom. Edges to $t_{N+1}$ are marked in bold when they belong to an attention head with top-10 function vector score for the task.

**Present-Past Task**

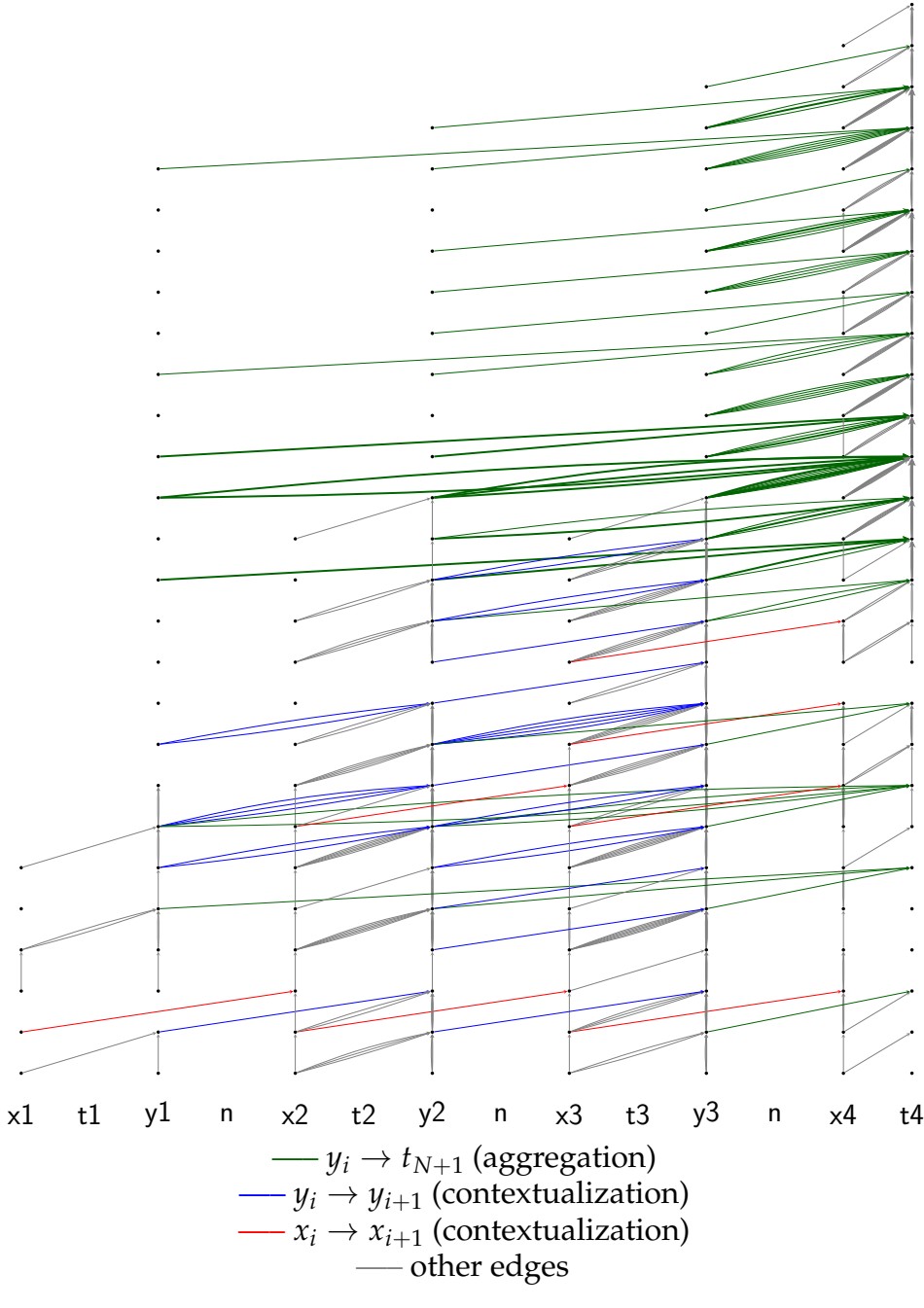

Figure 11: Activation-level circuit. Only attention edges are shown. Each row corresponds to one layer, starting from the input layer at the bottom. Edges to $t_{N+1}$ are marked in bold when they belong to an attention head with top-10 function vector score for the task.

**Person-Sport Task**

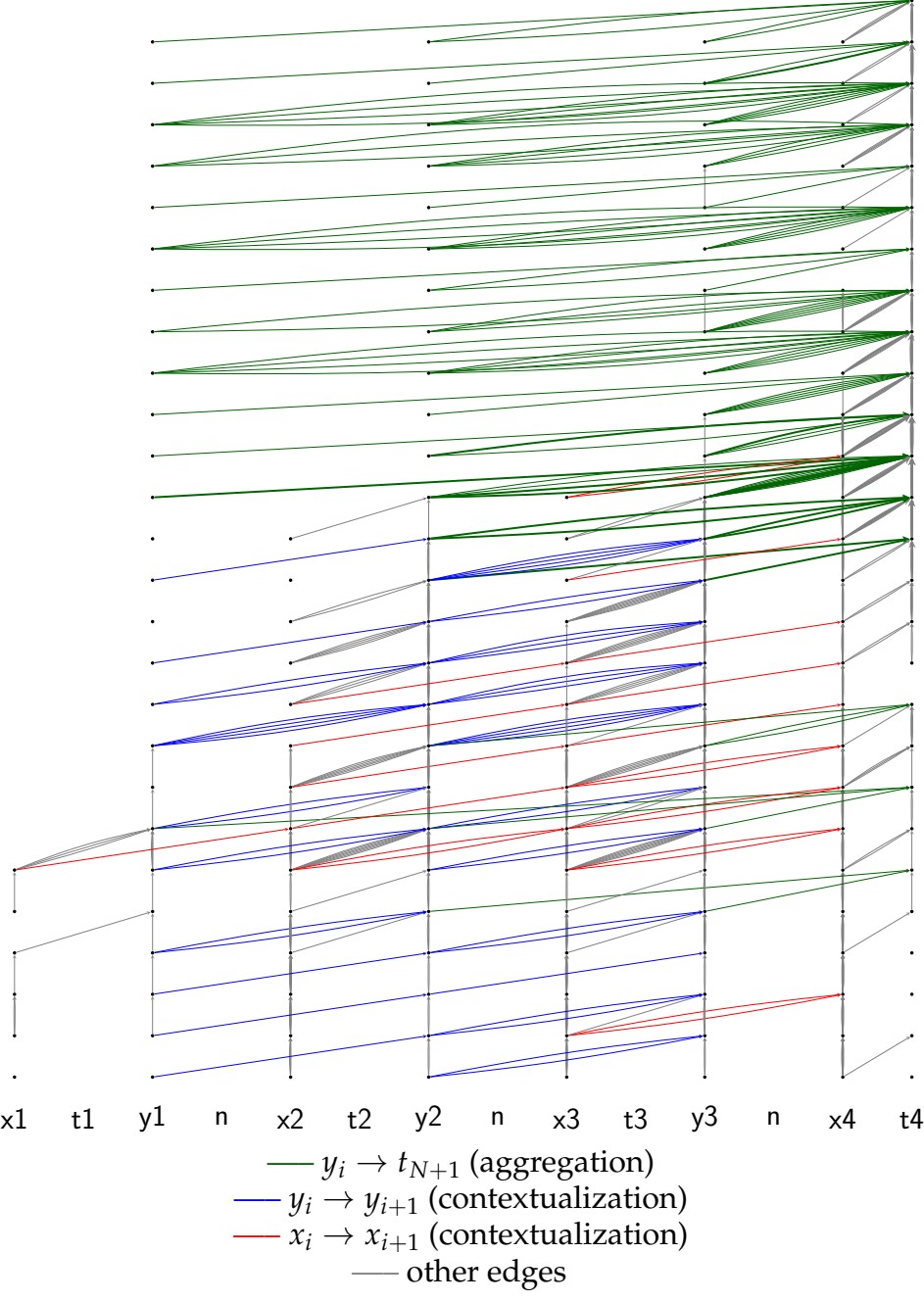

Figure 12: Activation-level circuit. Only attention edges are shown. Each row corresponds to one layer, starting from the input layer at the bottom. Edges to $t_{N+1}$ are marked in bold when they belong to an attention head with top-10 function vector score for the task.

**Copying Task**

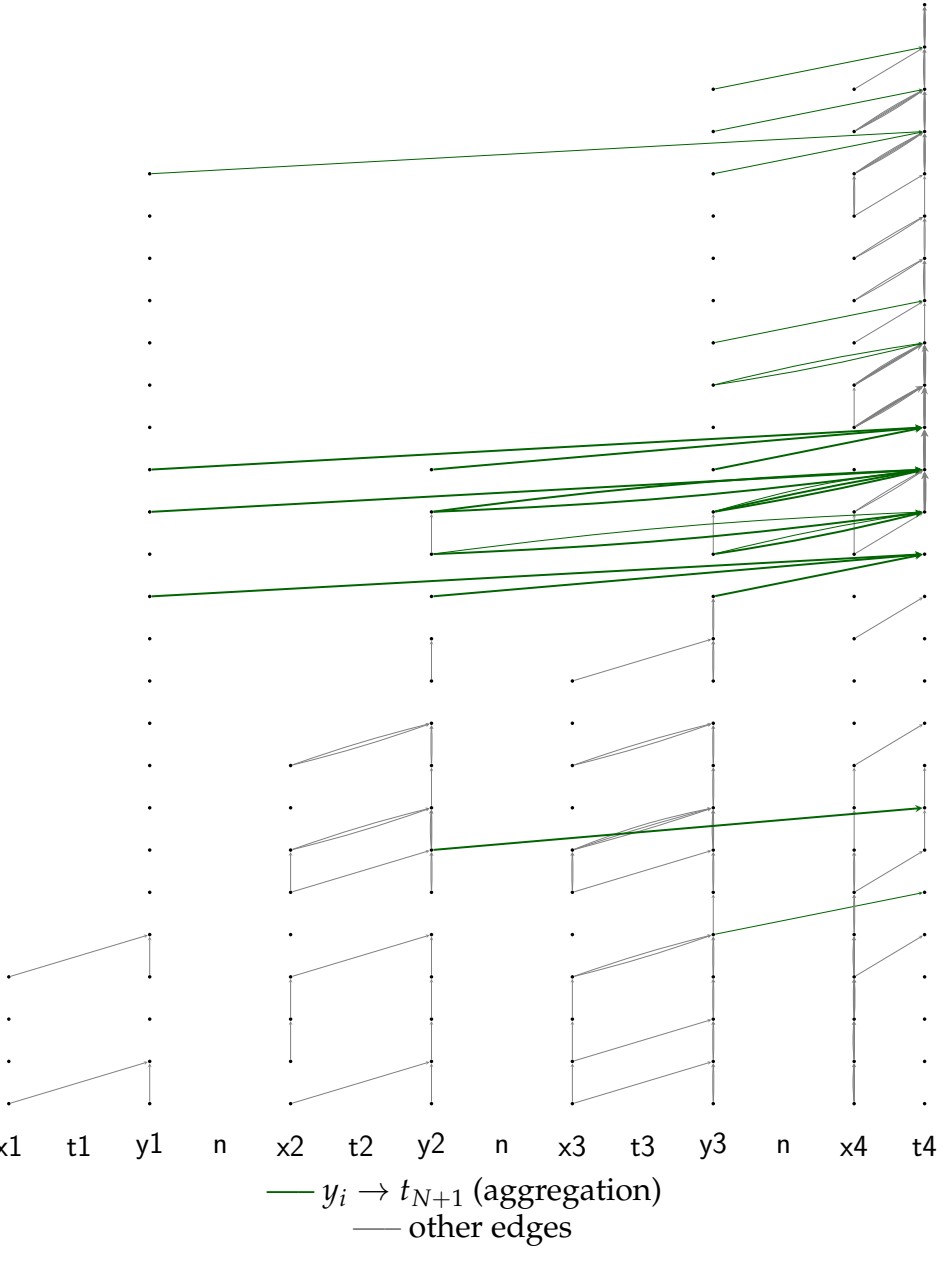

$\qquad y_i \rightarrow t_{N+1}$ (aggregation)

—— other edges

Figure 13: Activation-level circuit. Only attention edges are shown. Each row corresponds to one layer, starting from the input layer at the bottom. Edges to $t_{N+1}$ are marked in bold when they belong to an attention head with top-10 function vector score for the task.

**Capitalization (AGGREGATION Subcircuit)**

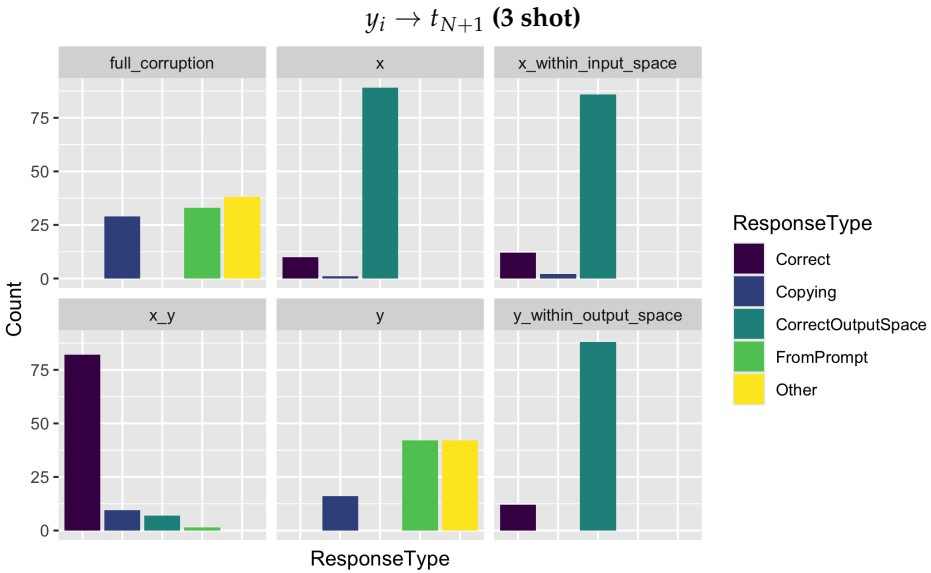

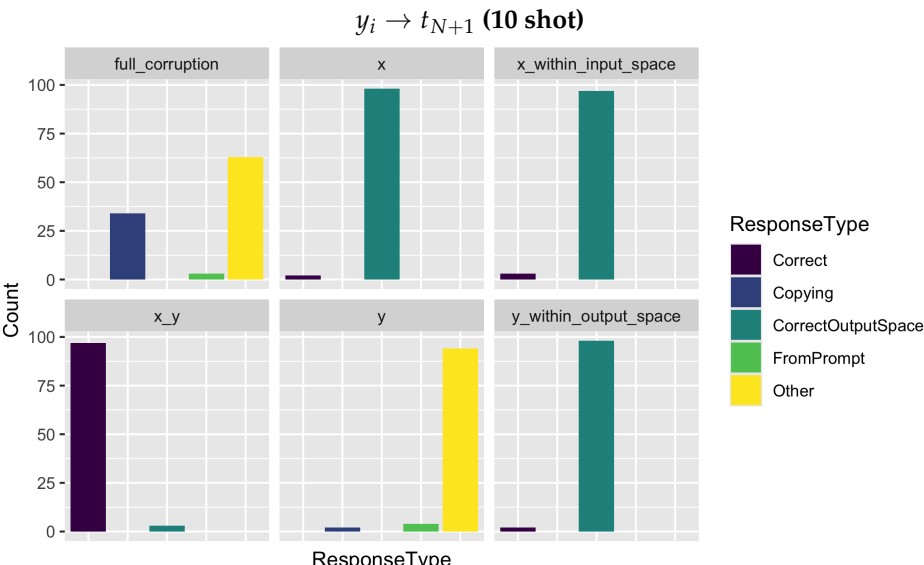

Figure 14: Error patterns: 10 shots (top) and 3 shots (bottom), $y_i \rightarrow t_{N+1}$(Capitalization). Perturbing the functional behavior but leaving the output space intact leads to incorrect responses staying in the correct output space (capitalized tokens; dark blue). Disrupting the output space leads to production of other tokens. See Appendix G for further information.

2. equals the query $x_{N+1}$, or

3. is from the tasks correct output space $\mathcal{Y}$ (e.g., a capitalized word, a city, etc – see Figure 25, or

4. matches a word from the corrupted prompt, and is thus likely copied from there, or

5. is any other kind of word

**Copying (AGGREGATION Subcircuit)**

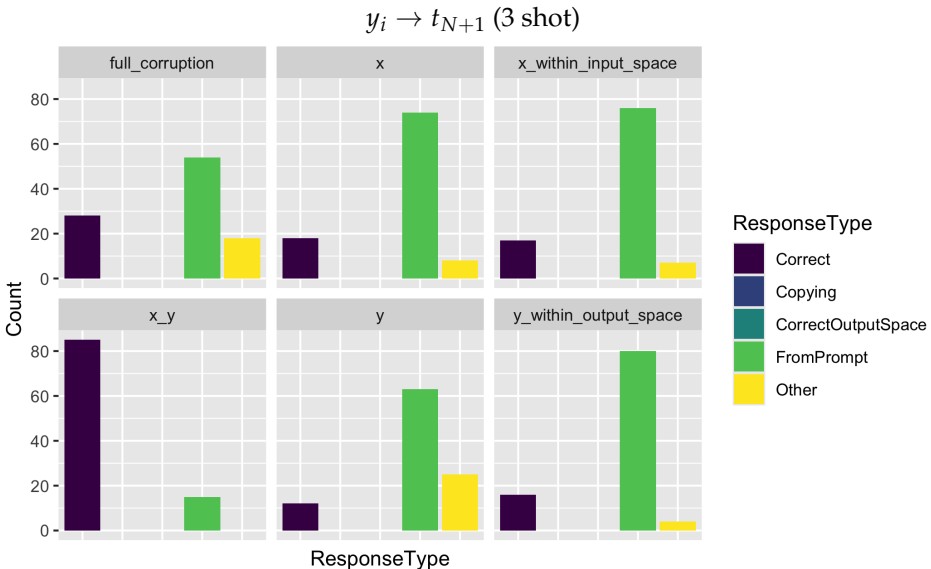

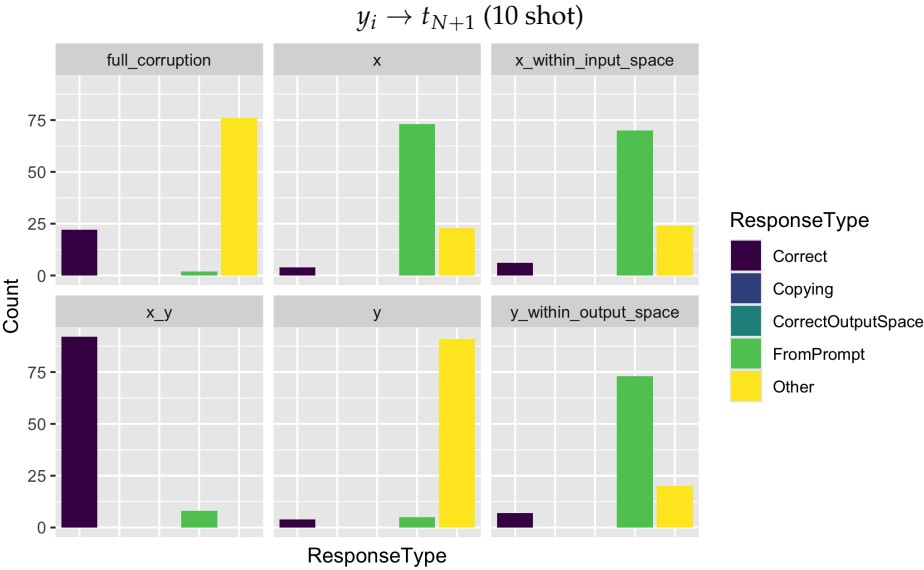

Figure 15: Error patterns for ablations in the (aggregation-only) circuit for Copying. Breaking the functional relationship between $x$ and $y$ often leads to reproduction of other tokens from the corrupted prompt (light green), or the production of other tokens (yellow). See Appendix G for further information.

**Country-Capital (AGGREGATION Subcircuit)**

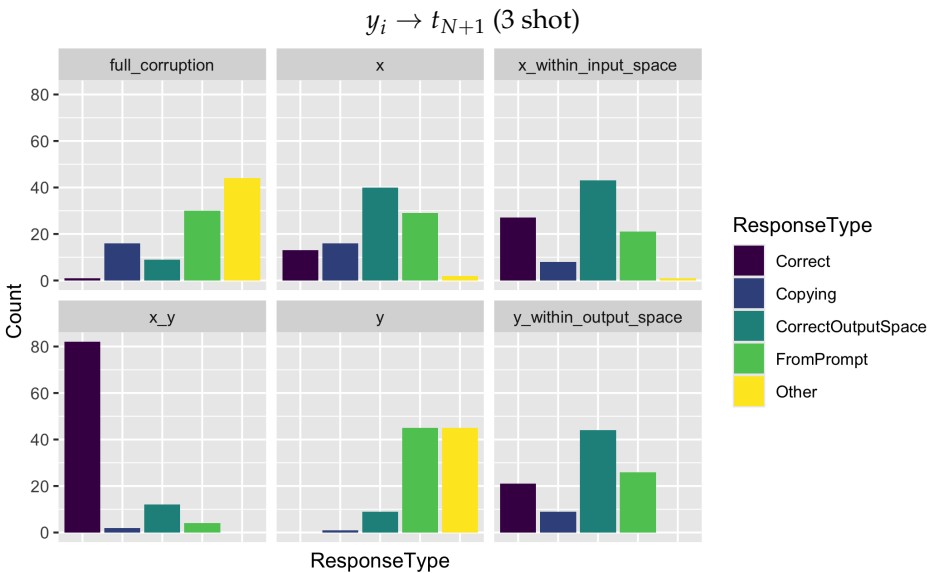

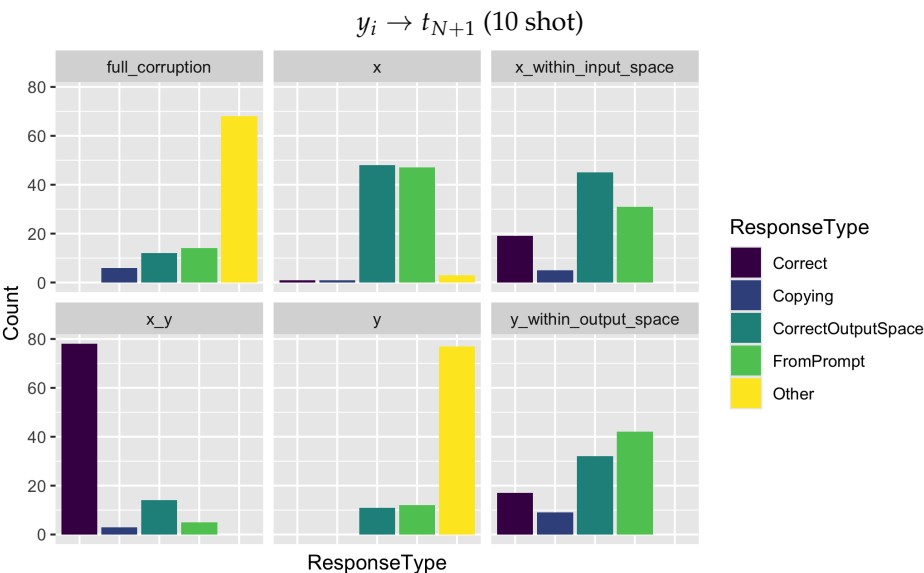

Figure 16: Error patterns for ablations in the aggregation edges for Country-Capital. See Appendix G for further information.

**Person-Sport (AGGREGATION Subcircuit)**

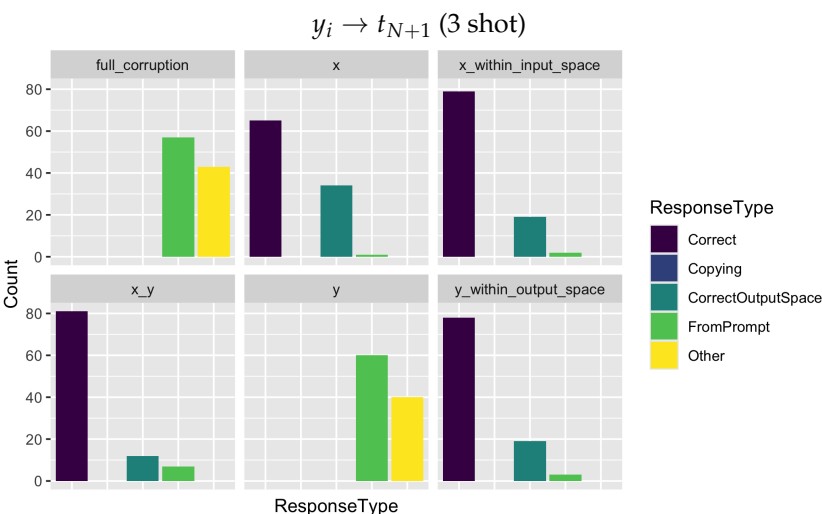

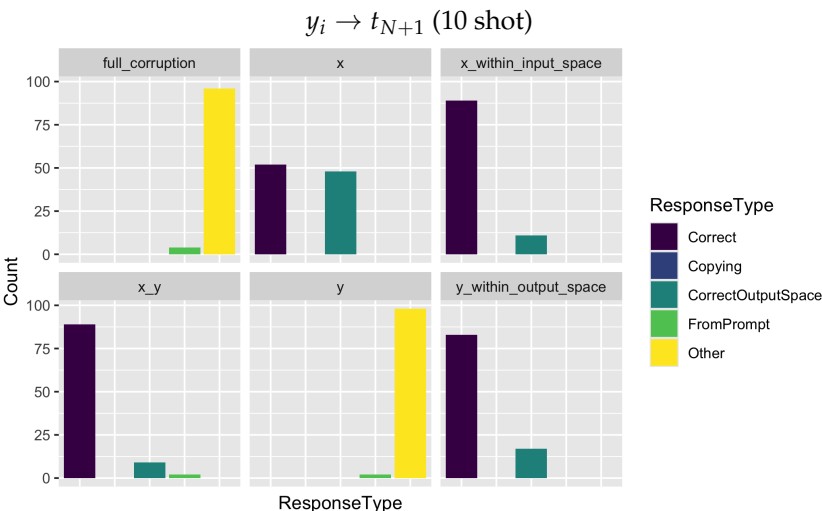

Figure 17: Error patterns in the Person-Sport task. The downstream effect of the aggregation edges suffers only weakly unless $x$ or $y$ leave their respective spaces. If $x$ leaves $\mathcal{X}$, predictions still are often correct or from the correct output space, showing substantial robustness in task inference. If $y$ leaves $\mathcal{Y}$, the model copies a token from the prompt or provides some other token. See Appendix G for further information.

**Present-Past (AGGREGATION Subcircuit)**

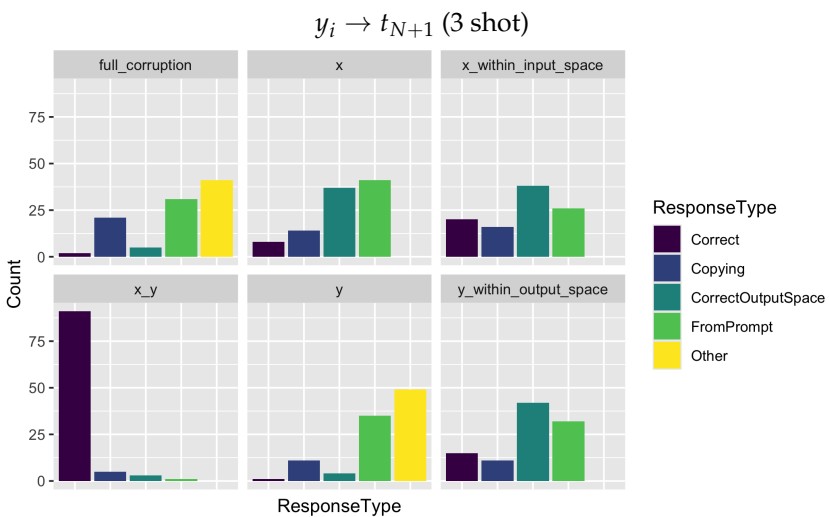

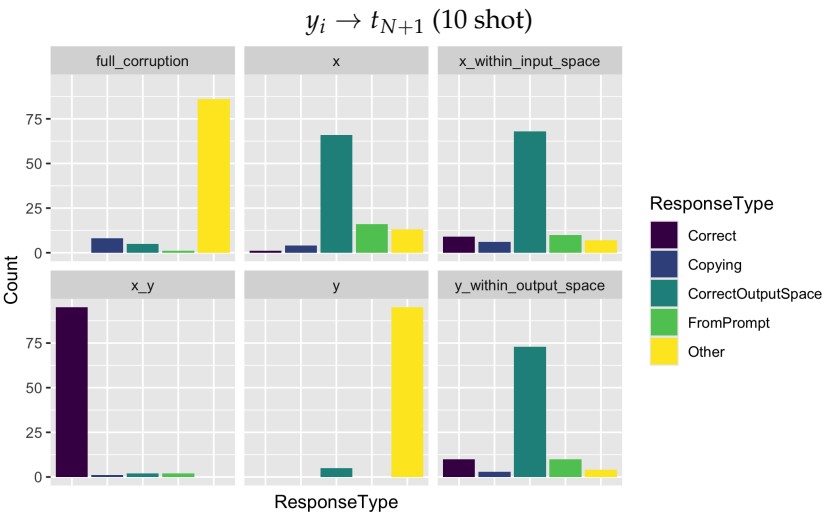

Figure 18: Present-Past task. Aggregation subcircuit. Disrupting the functional relation between $x$ and $y$, even if leaving the input- and output-spaces unchanged, leads to a large amount of reproduction from the correct output space (past-tense verbs), copying from the prompt, or other errors. See Appendix G for further information.

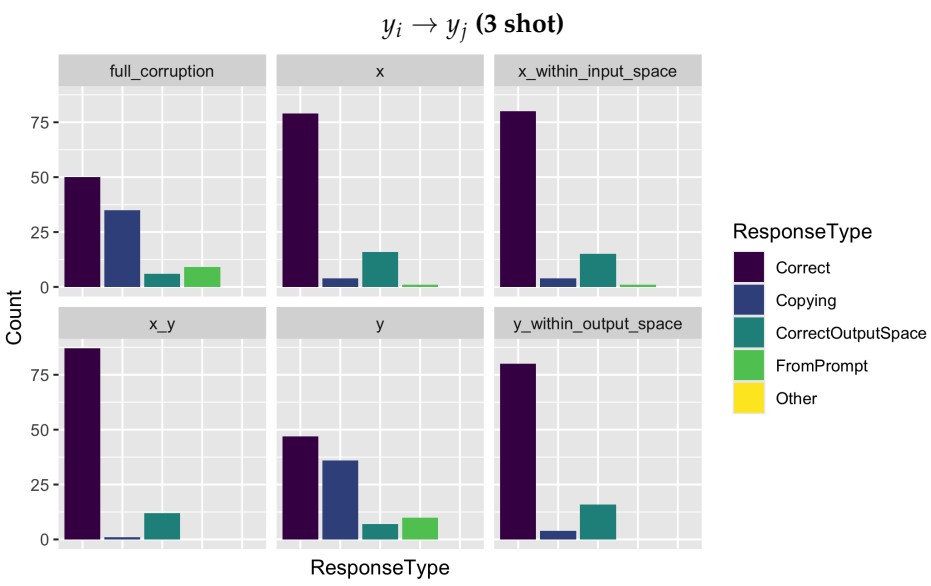

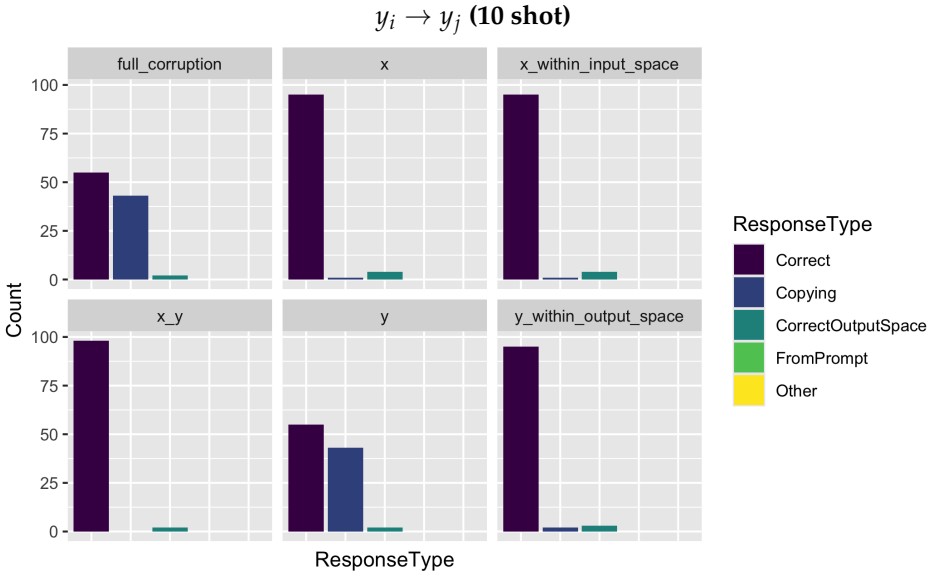

Figure 19: Error patterns for ablating $y_i \to y_j$ edges (Capitalization). The $y$-axis denotes the number of datapoints in each class. Ablating information about the output space leads to a large number of copy-type mistakes. See Appendix G for further information.

**Country-Capital (CONTEXTUALIZATION Subcircuit)**

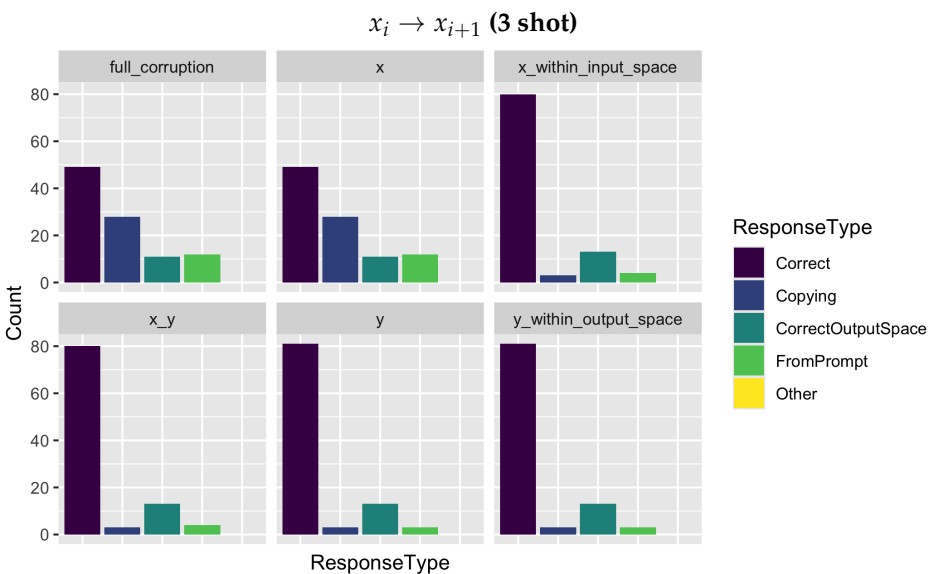

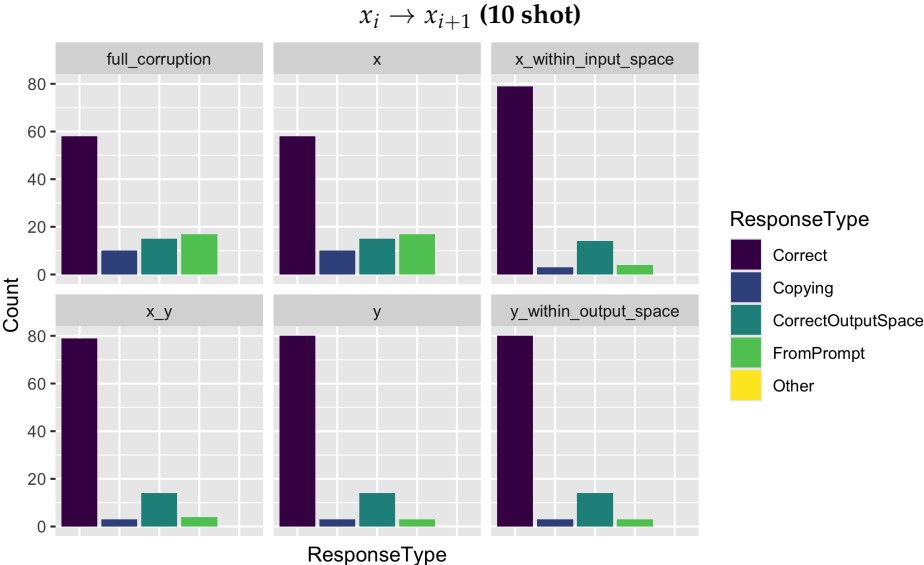

Figure 20: Error patterns for ablations in the $x_i \rightarrow x_{i+1}$ edges for Country-Capital. The $y$-axis denotes the number of datapoints in each class. Corrupting the input-space information in Neighboring-Xs edges leads to a substantial fraction of copying responses, i.e., the functional input-output behavior breaks down, in particular at 3 shots. See Appendix G for further information.

**Country-Capital (CONTEXTUALIZATION Subcircuit)**

$y_i \rightarrow y_{i+1}$ **(3 shot)**

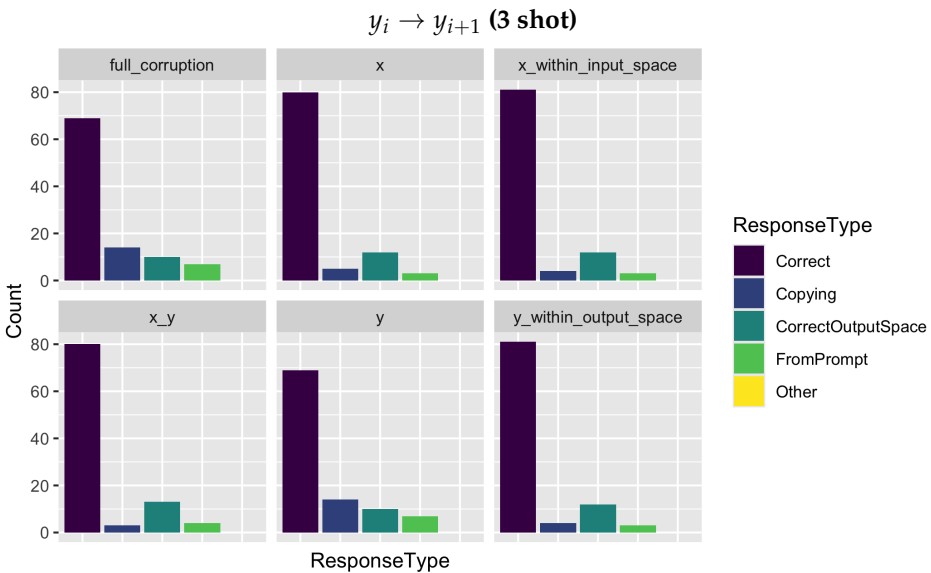

$y_i \rightarrow y_{i+1}$ **(10 shot)**

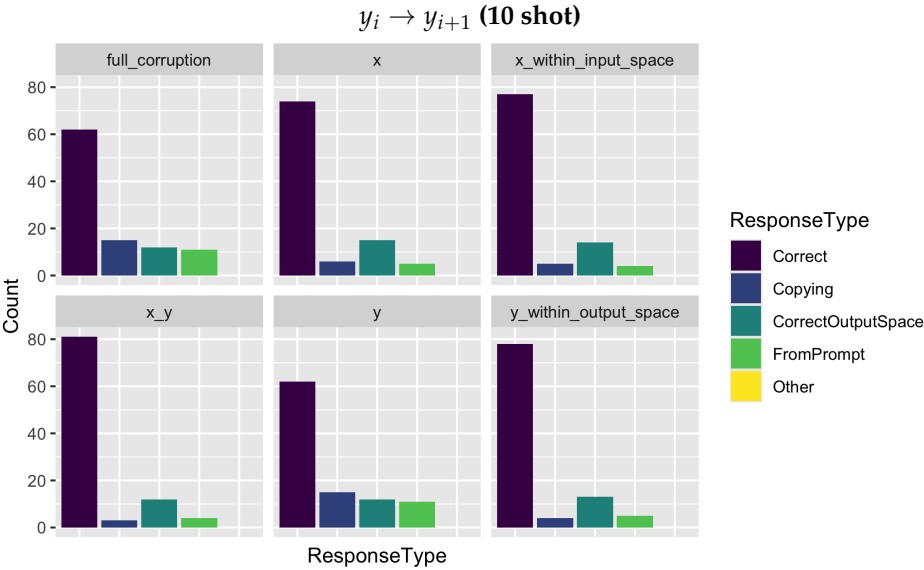

Figure 21: Error patterns for ablations in the $y_i \rightarrow y_{i+1}$ edges for Country-Capital. The $y$-axis denotes the number of datapoints in each class. Corrupting the output space information produces a (limited but statistically significant) increase in copying responses. See Appendix G for further information.

1. Original

```
Moldova\tChisinau\nGeorgia\tTbilisi\nTanzania\tDodoma\nGermany\t
```

2. Full ablation:

```
misreference\tnothosaurus\namide\tjerl\ncapes\tlinalool\ncapes\t
```

3. Ablating the inputs $x_i$ and the input space $\mathcal{X}$:

```
misreference\tChisinau\nwhere\tTbilisi\nany\tDodoma\nGermany\t
```

4. Ablating the inputs $x_i$, while preserving the input space $\mathcal{X}$:

```
Montenegro\tChisinau\nPanama\tTbilisi\nMalaysia\tDodoma\nGermany\t
```

5. Ablating the outputs $y_i$ and the output space $\mathcal{Y}$:

```
Moldova\tsaponify\nGeorgia\tbeeswings\nTanzania\tculverkey\nGermany\t
```

6. Ablating the outputs $y_i$, while preserving the output space $\mathcal{Y}$:

```
Moldova\tChisinau\nGeorgia\tZagreb\nTanzania\tDodoma\nGermany\t
```

7. Ablating examples, while preserving the functional relationship:

```
Montenegro\tPodgorica\nPanama\tPanama City\nMalaysia\tKuala Lumpur\nGermany\t
```

Figure 22: Examples for corrupted prompts used for patching, for the Country-Capital task (3-shot setting). We use full ablations (2) for identifying position-level circuits, and more refined ablations (3–7) for identifying the type of information flow routed between positions (Figure 3).

## H   More Details on Corrupted Prompts

See Figures 22 for examples of the different corrupted prompts used for patching.

An important constraint in choosing corrupted prompts is to keep the number of tokens in each $x_i$ and $y_i$ identical. For full corruption, the vocabulary is a subset of words used in the Capitalization task, which are random English words. Corrupted tokens could repeat in one prompt, but were never the same as the query token. For corruption within the input or output space, we use the input or output set of the training set of the corresponding task.

```
(1)  extend\textended\nset\tset\nwet\twet\nput\tput\nbid\tbid\nlast
     \tlasted\ncut\tcut\nspread\tspread\nreflect\treflected\nupset
     \tupset\nnotice\t

(2)  split\tsplit\nbid\tbid\nlet\tlet\nbroadcast\tbroadcast\nspread
     \tspread\nupset\tupset\nhit\thit\nset\tset\nupset\tupset\nupset
     \tupset\nnotice\t
```

Figure 23: (1) Example prompt in the ambiguity experiment, with 7 ambiguous and 3 unambiguous examples. (2) A prompt where *all* examples are ambiguous, used for testing the hypotheses H1 and H2.

| | Present-Past | | Person-Sport | | Capitalization | | Country-Capital | |
|---|---|---|---|---|---|---|---|---|
| | 3-shot | 10-shot | 3-shot | 10-shot | 3-shot | 10-shot | 3-shot | 10-shot |
| falcon3 | 0.90 | 0.95 | 0.46 | 0.60 | 0.99 | 1.00 | 0.69 | 0.69 |
| llama3 | 0.55 | 0.63 | 0.23 | 0.28 | 0.53 | 0.71 | 0.30 | 0.41 |
| phi2 | 0.96 | 0.98 | 0.69 | 0.83 | 0.94 | 0.98 | 0.87 | 0.88 |
| qwen2 | 0.43 | 0.59 | 0.28 | 0.27 | 0.63 | 0.66 | 0.41 | 0.45 |
| qwen2-3b | 0.51 | 0.58 | 0.19 | 0.20 | 0.57 | 0.63 | 0.43 | 0.42 |
| smollm2 | 0.95 | 0.97 | 0.67 | 0.71 | 1.00 | 1.00 | 0.89 | 0.91 |

Table 7: Accuracies of six other models at the 2B/3B scale. On average across the four tasks, each model overall underperforms Gemma-2 2B at 10 shots on these tasks (Table 2), further motivating focusing on Gemma-2.

| | | Country-Capital | |
|---|---|---|---|
| | | 3-shot | 10-shot |
| $x_{N+1} \to t_{N+1}$ | all $x_i$ with random words | 0.58 | 0.73 |
| $y_i \to t_{N+1}$ | $x_j, j < i$ with random word | 0.70 | 0.65 |
| $x_{N+1} \to t_{N+1}$ | all $x_i$ within input space | 0.80 | 0.80 |
| $y_i \to t_{N+1}$ | $x_j, j < i$ within input space | 0.78 | 0.77 |

Table 8: Determining through which downstream paths $x_i \to x_j$ edges provides information: A priori, given the position-level circuit for the Country-Capital task (Figure 24), $x_i \to x_j$ edges might provide information affecting the final prediction both via $x_i \to x_{i+1} \to y_{i+1} \to t_{N+1}$ and via $x_N \to x_{N+1} \to t_{N+1}$. Here, we patch with prior $x_i$'s, for both types of donwstream edges ($x_{N+1} \to t_{N+1}$ and $y_i \to t_{N+1}$), on the Country-Capital task. We consider both patching with random words (eliminating information both about $x_i$ and $\mathcal{X}$), and patching with other words in the input space (eliminating information about $x_i$ but not $\mathcal{X}$). Given the position-level circuit for the Country-Capital task (Figure 24), information about the manipulated words can flow into these edges only via $x_i \to x_{i+1}$ connections. Accuracy drops substantively compared to the full circuit (0.79 at 3 shots, 0.8 at 10 shots) when eliminating information about $\mathcal{X}$, both when applying this patch to $x_{N+1} \to t_{N+1}$ or to $y_i \to t_{N+1}$. This shows that $x_i \to x_{i+1}$ edges contextualize both the individual few-shot examples and the query $x_{N+1}$ with information about the input space $\mathcal{X}$.

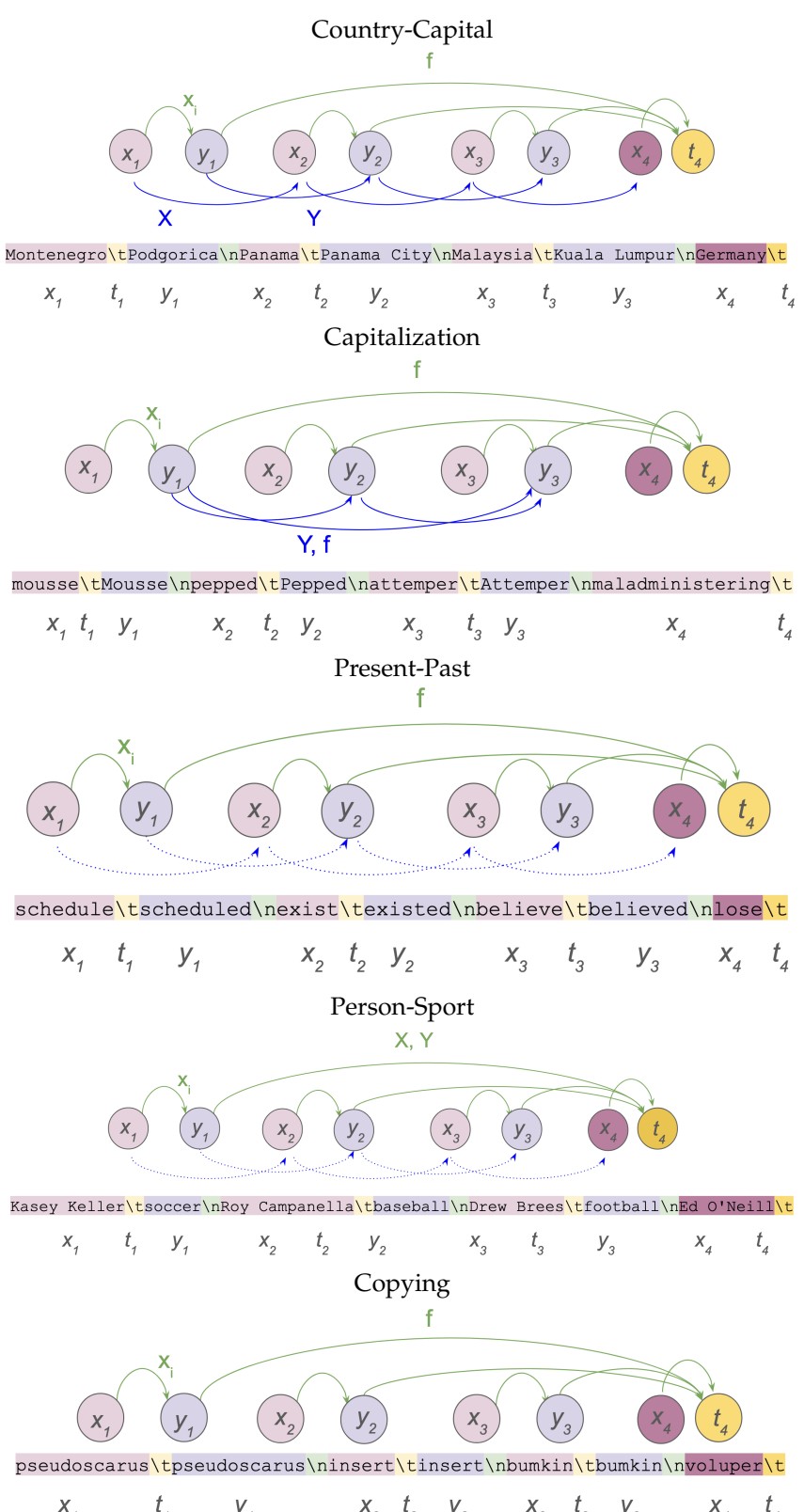

Figure 24: Position-level circuits for 3-shot prompts; edges are annotated for information as identified in our patching experiments (subsection 3.2). Dotted connections are part of the selected circuits, but the accuracy drop associated with ablating them is not statistically significant.

| Task | Example | $\mathcal{X}$ | $\mathcal{Y}$ |
|---|---|---|---|
| Copying | f(pseudoscarus) = pseudoscarus | arbitrary words | arbitrary words |
| Capitalization | f(mousse) = Mousse | lowercase words | uppercase words |
| Country-Capital | f(Malaysia) = Kuala Lumpur | countries | capitals |
| Present-Past | f(give) = gave | verbs | past tense verbs |
| Person-Sport | f(Kasey Keller) = soccer | sportspeople | sports |

Figure 25: Input spaces, output spaces, and functions $f : \mathcal{X} \rightarrow \mathcal{Y}$ for the five tasks considered in this paper.

| | | FULL | AGGREGATION $+ x_i \rightarrow x_{i+1}$ $+ y_i \rightarrow y_{i+1}$ | REMOVE-SEPS | AGGREGATION |
|---|---|---|---|---|---|
| 3-shot | no ambiguous | 0.96 | 0.93 | 0.92 | 0.84 |
| | 1 ambiguous | 0.98 | 0.73 | 0.81 | 0.66 |
| | 2 ambiguous | 0.66 | 0.41 | 0.47 | 0.34 |
| 10-shot | no ambiguous | 0.99 | 0.94 | 0.92 | 0.93 |
| | 3 ambiguous | 0.98 | 0.98 | 0.98 | 0.95 |
| | 5 ambiguous | 0.98 | 0.93 | 0.98 | 0.80 |
| | 7 ambiguous | 0.95 | 0.68 | 0.85 | 0.56 |

Table 9: Accuracy of the Present-Past task in the presence of different numbers of ambiguous few-shot examples (Section 3.3). In the presence of ambiguity, the full model (FULL) continues to perform well even when 7 out of 10 examples are ambiguous between the Present-Past and Copying tasks. In contrast, the accuracy of the AGGREGATION circuit drops to 56% as the number of ambiguous examples increases from 0 to 7. Adding contextualization between $x_i$'s an $y_i$'s helps, but does not close the gap to the full model; indeed, even ablating edges involving separators preceding $t_{N+1}$ hurts (REMOVE-SEPS), in contrast to the standard test sets.

**Country-Capital**

| Layer | Head | Edges |
|---|---|---|
| 12 | 1 | $t_4 \to t_4$, $x_4 \to x_4$, $x_3 \to x_4$, $\mathbf{y_2} \to \mathbf{t_4}$, $y_2 \to y_2$, $y_2 \to y_3$, $\mathbf{y_3} \to \mathbf{t_4}$, $\mathbf{y_1} \to \mathbf{t_4}$, $y_3 \to y_3$, $y_1 \to y_2$ |
| 14 | 0 | $t_4 \to t_4$, $x_4 \to x_4$, $y_2 \to y_2$, $\mathbf{y_2} \to \mathbf{t_4}$, $\mathbf{y_3} \to \mathbf{t_4}$, $y_3 \to y_3$, $\mathbf{y_1} \to \mathbf{t_4}$ |
| 14 | 1 | $x_4 \to t_4$, $t_4 \to t_4$, $x_4 \to x_4$, $\mathbf{y_2} \to \mathbf{t_4}$, $\mathbf{y_3} \to \mathbf{t_4}$, $y_3 \to y_3$, $\mathbf{y_1} \to \mathbf{t_4}$ |
| 14 | 3 | $x_4 \to t_4$, $t_4 \to t_4$, $x_4 \to x_4$, $x_3 \to x_4$, $\mathbf{y_2} \to \mathbf{t_4}$, $\mathbf{y_3} \to \mathbf{t_4}$, $y_3 \to y_3$, $\mathbf{y_1} \to \mathbf{t_4}$ |
| 13 | 4 | $x_4 \to t_4$, $t_4 \to t_4$, $x_4 \to x_4$, $x_3 \to x_4$, $\mathbf{y_2} \to \mathbf{t_4}$, $y_2 \to y_2$, $y_2 \to y_3$, $\mathbf{y_3} \to \mathbf{t_4}$, $\mathbf{y_1} \to \mathbf{t_4}$, $y_3 \to y_3$ |
| 15 | 7 | $x_4 \to t_4$, $t_4 \to t_4$, $x_4 \to x_4$, $x_3 \to x_4$, $\mathbf{y_2} \to \mathbf{t_4}$, $\mathbf{y_3} \to \mathbf{t_4}$, $y_3 \to y_3$, $\mathbf{y_1} \to \mathbf{t_4}$ |
| 13 | 7 | $x_4 \to t_4$, $t_4 \to t_4$, $x_4 \to x_4$, $y_2 \to y_2$, $\mathbf{y_3} \to \mathbf{t_4}$, $y_3 \to y_3$, $\mathbf{y_1} \to \mathbf{t_4}$ |
| 17 | 7 | $t_4 \to t_4$, $x_4 \to x_4$, $\mathbf{y_3} \to \mathbf{t_4}$, $x_4 \to t_4$ |
| 16 | 6 | $t_4 \to t_4$, $\mathbf{y_3} \to \mathbf{t_4}$, $\mathbf{y_2} \to \mathbf{t_4}$, $x_4 \to t_4$ |
| 15 | 3 | $x_4 \to t_4$, $t_4 \to t_4$, $x_4 \to x_4$, $\mathbf{y_3} \to \mathbf{t_4}$, $y_2 \to y_2$, $y_3 \to y_3$ |

**Capitalization**

| Layer | Head | Edges |
|---|---|---|
| 14 | 0 | $\mathbf{y_2} \to \mathbf{t_4}$, $\mathbf{y_1} \to \mathbf{t_4}$, $\mathbf{y_3} \to \mathbf{t_4}$, $t_4 \to t_4$ |
| 14 | 1 | $\mathbf{y_2} \to \mathbf{t_4}$, $\mathbf{y_1} \to \mathbf{t_4}$, $\mathbf{y_3} \to \mathbf{t_4}$, $t_4 \to t_4$ |
| 12 | 1 | $\mathbf{y_2} \to \mathbf{t_4}$, $\mathbf{y_1} \to \mathbf{t_4}$, $\mathbf{y_3} \to \mathbf{t_4}$ |
| 13 | 4 | $\mathbf{y_2} \to \mathbf{t_4}$, $y_2 \to y_3$, $y_2 \to y_2$, $y_3 \to y_3$, $\mathbf{y_3} \to \mathbf{t_4}$ |
| 15 | 3 | $t_4 \to t_4$ |
| 17 | 7 | $t_4 \to t_4$ |
| 15 | 0 | $\mathbf{y_2} \to \mathbf{t_4}$, $\mathbf{y_1} \to \mathbf{t_4}$, $\mathbf{y_3} \to \mathbf{t_4}$, $t_4 \to t_4$ |
| 17 | 3 | $t_4 \to t_4$, $x_4 \to t_4$ |
| 16 | 6 | $t_4 \to t_4$, $x_4 \to t_4$ |
| 19 | 5 | $\mathbf{y_2} \to \mathbf{t_4}$, $\mathbf{y_3} \to \mathbf{t_4}$, $t_4 \to t_4$ |

**Present-Past**

| Layer | Head | Edges |
|---|---|---|
| 14 | 0 | $\mathbf{y_3} \to \mathbf{t_4}$, $\mathbf{y_2} \to \mathbf{t_4}$, $t_4 \to t_4$, $\mathbf{y_1} \to \mathbf{t_4}$ |
| 14 | 1 | $\mathbf{y_3} \to \mathbf{t_4}$, $\mathbf{y_2} \to \mathbf{t_4}$, $t_4 \to t_4$, $\mathbf{y_1} \to \mathbf{t_4}$ |
| 12 | 1 | $\mathbf{y_3} \to \mathbf{t_4}$, $y_2 \to y_3$, $t_4 \to t_4$, $\mathbf{y_2} \to \mathbf{t_4}$, $y_3 \to y_3$, $\mathbf{y_1} \to \mathbf{t_4}$ |
| 15 | 0 | $\mathbf{y_3} \to \mathbf{t_4}$, $\mathbf{y_2} \to \mathbf{t_4}$, $t_4 \to t_4$, $\mathbf{y_1} \to \mathbf{t_4}$ |
| 13 | 4 | $\mathbf{y_3} \to \mathbf{t_4}$, $\mathbf{y_2} \to \mathbf{t_4}$, $t_4 \to t_4$ |
| 15 | 3 | $t_4 \to t_4$ |
| 23 | 5 | $\mathbf{y_3} \to \mathbf{t_4}$, $t_4 \to t_4$, $x_4 \to t_4$ |
| 14 | 7 | $\mathbf{y_3} \to \mathbf{t_4}$, $\mathbf{y_2} \to \mathbf{t_4}$, $t_4 \to t_4$, $x_4 \to t_4$ |
| 20 | 6 | $x_4 \to t_4$, $t_4 \to t_4$, $x_4 \to x_4$, $\mathbf{y_3} \to \mathbf{t_4}$ |
| 23 | 0 | $t_4 \to t_4$ |

Figure 26: Top-10 function vector heads on each task, and their roles in the activation-level circuits (3-shot prompts, $N = 3$). Edges from the form $y_i \to t_{N+1}$ are highlighted in boldface. Most are involved in at least one such edge, showing that function vector heads are causally involved in the aggregation of task information from few-shot examples. Many edges also causally participate in $t_{N+1} \to t_{N+1}$ edges, suggesting processing or forwarding of task information. Heads also sometimes participare in edges not going to $t_{N+1}$ (those do not enter the function vector score calculation); these are also shown here. For the two remaining tasks, see next figure.

**Copying**

| Layer | Head | Edges |
|---|---|---|
| 14 | 0 | $\mathbf{y_3} \to \mathbf{t_4}, \mathbf{y_2} \to \mathbf{t_4}, t_4 \to t_4, \mathbf{y_1} \to \mathbf{t_4}$ |
| 14 | 1 | $\mathbf{y_3} \to \mathbf{t_4}, \mathbf{y_2} \to \mathbf{t_4}, t_4 \to t_4$ |
| 12 | 1 | $\mathbf{y_3} \to \mathbf{t_4}, \mathbf{y_2} \to \mathbf{t_4}, \mathbf{y_1} \to \mathbf{t_4}$ |
| 13 | 4 | $\mathbf{y_3} \to \mathbf{t_4}, \mathbf{y_2} \to \mathbf{t_4}$ |
| 15 | 3 | $t_4 \to t_4$ |
| 17 | 3 | $x_4 \to t_4, t_4 \to t_4$ |
| 15 | 0 | $\mathbf{y_3} \to \mathbf{t_4}, \mathbf{y_2} \to \mathbf{t_4}, t_4 \to t_4, \mathbf{y_1} \to \mathbf{t_4}$ |
| 17 | 7 | |
| 6 | 4 | $\mathbf{y_2} \to \mathbf{t_4}$ |
| 16 | 6 | $x_4 \to t_4, t_4 \to t_4$ |

**Person-Sport**

| Layer | Head | Edges |
|---|---|---|
| 12 | 1 | $\mathbf{y_2} \to \mathbf{t_4}, \mathbf{y_3} \to \mathbf{t_4}, t_4 \to t_4, y_2 \to y_3, x_4 \to x_4, y_3 \to y_3$ |
| 15 | 7 | $\mathbf{y_2} \to \mathbf{t_4}, \mathbf{y_3} \to \mathbf{t_4}, x_4 \to t_4, t_4 \to t_4, x_4 \to x_4$ |
| 14 | 1 | $t_4 \to t_4, \mathbf{y_3} \to \mathbf{t_4}, \mathbf{y_2} \to \mathbf{t_4}$ |
| 14 | 0 | $x_3 \to x_4, \mathbf{y_1} \to \mathbf{t_4}, \mathbf{y_2} \to \mathbf{t_4}, \mathbf{y_3} \to \mathbf{t_4}, t_4 \to t_4, x_4 \to x_4$ |
| 13 | 5 | $\mathbf{y_2} \to \mathbf{t_4}, \mathbf{y_3} \to \mathbf{t_4}, x_4 \to t_4, t_4 \to t_4, x_4 \to x_4, y_3 \to y_3$ |
| 14 | 4 | $t_4 \to t_4, \mathbf{y_3} \to \mathbf{t_4}, x_4 \to x_4, x_4 \to t_4$ |
| 17 | 7 | $t_4 \to t_4, x_4 \to t_4$ |
| 13 | 4 | $t_4 \to t_4, \mathbf{y_3} \to \mathbf{t_4}, x_4 \to x_4, \mathbf{y_2} \to \mathbf{t_4}$ |
| 24 | 3 | $t_4 \to t_4, \mathbf{y_3} \to \mathbf{t_4}$ |
| 15 | 3 | $t_4 \to t_4, x_4 \to x_4, x_4 \to t_4$ |

Figure 27: Continuation of previous figure.

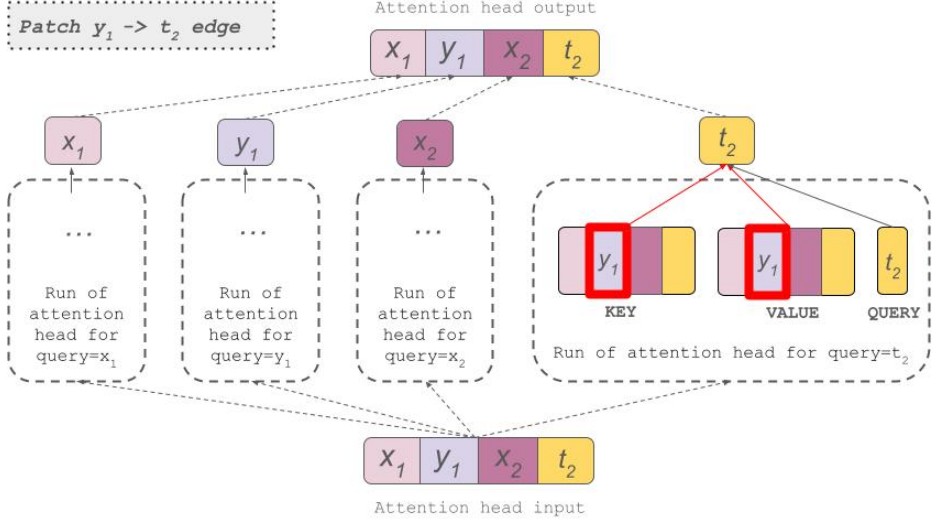

Figure 28: Patching the edge $y_1 \to t_2$ inside an attention head. We compute attention separately for queries at each position. For the ablation, we modify only the computation at $t_2$ query position, replacing the $y_1$ activation in the K and V matrices with its counterpart computed on a corrupted input prompt. This ensures $t_2$ cannot access information unique to the clean $y_1$ activation, at least not directly via the $y_1 \to t_2$ edge.

