# OpenReview forum: "Contextualize-then-Aggregate: Circuits for In-Context Learning in Gemma-2 2B"
_colmweb.org/COLM/2025/Conference — COLM 2025_

### Official Review · Reviewer_mZBA · 2025-05-07

**Rating:** 7
**Confidence:** 4
**Ethics Flag:** 1

**Summary:**

This paper investigates few-shot ICL prediction in transformer LLMs (Gemma 2 family) using position-level circuits. In their analysis of information flow, they find that there are two main steps models use to infer task-specific information from few-shot examples: contextualization and aggregation. While some of the findings were already known/hypothesized in previous work, I do like the overall contributions of this work. The most interesting finding to me was that label tokens of few-shot examples are contextualized both by their corresponding inputs and by previous examples' labels, which allows for disambiguation even when some examples are unhelpful.

Overall, the findings of this paper are interesting, but the writing can be improved by making experimental details more clear and more explicitly noting the work's contributions from the beginning. With regard to clarity, a lot of the current writing assumes familiarity with recent interpretability and circuit-discovery work which may limit the current draft's effective reach. I think this can be overcome with some effort to clarify the main contributions of the work and provide context details where appropriate (either in main text or in appendix). I have tried to provide feedback in the following sections related to these main points. Despite this, I felt the flow of the paper was pretty good and the _contextualization_ of the nuanced contributions of this work with respect to previous work was nice and that its does help improve our understanding of how few-shot ICL prompting works a little better than before.

A separate quick note: This submission does not have numbered lines (so I wonder if the template used for submission is slightly incorrect?). I will use page number, section number, and quotes where helpful, to refer to relevant places in the paper.

**Questions To Authors:**

**Questions**
- (1) On page 7, you mention that the few-shot ICL circuit is not a "classical induction circuit" (paragraph: "Is the ICL aggregation circuit a type of induction circuit?"). While it may not copy literal tokens, do you think it could still be thought of as an "induction circuit" that "copies" forward a contextualized abstraction of the task (along with some other information)? This idea is suggested by Yang, et al. [4] and also seems related to the more abstract kind of induction described in Feucht, et al. [5] as well. (I'm aware these works are very recent - so feel free to ignore, but just curious on your thoughts here.)

- (2) Do you think the phenomenon observed w/ person-sport has to do with the size of the output space ($|\mathcal{Y}|$)? For example, the number of sports is very small in comparison to the number of verbs with past tenses. Sentiment tasks are similar in this regard (the label space is small - e.g., $\mathcal{Y}=$ {"positive", "negative"}) and I wonder if the kinds of subcircuits you're finding & information present causally relevant during the contextualization & aggregation steps depends on the size of the input and output spaces. Have you tested a single task (maybe synthetic) while you vary the size of the output space?

- (3) ("The edges causally transport only task information", page 6) Which edges is this referring to? Is it all of the parallel-circuit edges ($x_i \rightarrow y_i$, $x_{N+1} \rightarrow t_{N+1}$, and $y_i \rightarrow t_{N+1}$? Or only some subset of them?

___
**Suggestions**

The current text relies too heavily on assuming the reader is familiarity with current interpretability research (particular circuit discovery methods) to understand it, and this undersells the nice findings presented in the paper. I suggest you try and present some more background on your experimental design and process to provide some intuition for those who may be less familiar with the process of "circuit-finding". I would also suggest to promote your "sub-findings" in section 3 (in boxes) earlier. Some form of these findings should probably be highlighted in the introduction as contributions as this will clarify what you found and why it's different from previous work a bit more than what is currently presented as the two "main contributions".
    - For example, this might look like spelling out the findings from your analysis of the information content in the aggregation and contextualization subcircuits in the introduction: e.g.,  (a) the aggregation subcircuit mostly transports function-specific information, with some token-type information that is relevant for some tasks (e.g. person-sport), and (b) the contextualization subcircuit mostly transports information about the input space between $x_i$'s, and information about the output space and some function-specific information between the $y_i$'s, etc.
___

[4] Yang, et al. Emergent Symbolic Mechanisms Support Abstract Reasoning in Large Language Models, arxiv 2025. (https://arxiv.org/pdf/2502.20332)

[5] Feucht, et al. The Dual Route Model of Induction. arxiv 2025. (https://arxiv.org/pdf/2504.03022)

**Reasons To Accept:**

- The clarification of how task-specific representations are formed during few-shot ICL will likely be of interest to the broader LLM community, so this paper investigates an interesting question.

- The contributions of this paper present some nuanced differences from previous related work (Kharlapenko, et al. [a] , Cho, et al. [b], Todd, et al. [c]) and the paper is well-situated among existing literature. Specifically, the description of a contextualization step, in addition to an aggregation step, appears to capture more of the model's performance for few-shot ICL prompts than previous explanations, though some results are similar and were known or alluded to previously (mostly regarding the aggregation step).

- The disambiguation results for the contextualization subcircuit are a nice finding, and I think others would be interested in this as an explanation for why flipping labels for some tasks may not hurt performance as much as others. This should maybe be highlighted a bit more (in addition to the other sub-results - see suggestions section).

- Additionally, showing that the same proposed subcircuits explain the synthetic setting of ICL regression well is a nice data point in favor of the proposed framework for understanding few-shot ICL.

- Finally, the authors have included empirical results across model sizes (albeit of the same model family), and the results seem consistent which is nice.

___
[a] Kharlapenko, et al.  Scaling sparse feature circuit finding for in-context learning. arxiv 2025. (https://arxiv.org/abs/2504.13756)

[b] Cho, et al. Revisiting in-context learning inference circuit in large language models. ICLR 2025. (https://openreview.net/forum?id=xizpnYNvQq)

[c] Todd, et al. Function Vectors in Large Language Models. ICLR 2024 (https://openreview.net/forum?id=AwyxtyMwaG)

**Reasons To Reject:**

**Presentation and Clarity**

There are some assumptions in the writing that should be clarified.
- The first is that the kind of in-context learning investigated in this paper is few-shot ICL prompting. While many works take this as standard the definition for ICL, it's helpful to clarify. For example, see Yin, et al. [1] or Lampinen, et al. [2].

- Similarly, the writing assumes familiarity with a lot of previous interpretability research and circuit-discovery work. I'm not sure it's suitable for a broader audience in its current state without some additional clarification. This is one of the main reasons I've given a lower score, but I'd be willing to raise my score as writing becomes crisper and details become easier to follow along with.
    - For example, in the introduction the first time "circuit" is mentioned, its definition is already assumed to be known: "However, the circuit by which these vectors are assembled remains only partly understood" (page 1).  And later in the introduction, the lines "We use causal interventions to identify information flow that recovers at least 90% of the model’s performance. A key idea is to first identify a computation graph between the tokens in a prompt (Figure 1)" (page 1) give only a vague definition of what a circuit might be.  (It could be that "causally faithful information flow" is the same as a "circuit" (others might debate this), but this is never stated). Giving a clearer definition of what you mean by "circuit" would probably be helpful, and might help to set the stage better for what you're doing.
    - In section 3.1 "patching methodology", quite a few experimental details about the circuit-finding method and context are missing (things that someone familiar with circuit-discovery literature might assume are happening under the hood, but are not clear from the current text). These include things like why patching is being done (what it tells you), what the purpose of the counterfactual prompts is, what you're measuring when doing patching between counterfactual and standard prompts (change in logits vs probabilities), etc. Related to your specific setup, explaining in more detail how information across layers is treated for a single position would be helpful because this seems to be a key design choice of yours in finding a "position-level circuit". It might also be helpful to mention how many edges your circuit nodes can have between token positions. For example, are all connections in a layer counted as a single edge or as multiple edges in your position-level circuit?

Additional notes about things that might help improve the clarity of the presentation:
- The "parallel circuit" described in the paper appears to go by many names ("parallel circuit", "parallel subcircuit", "aggregation circuit", "aggregation-only circuit", etc.) and clarifying the differences between these, or reducing it down to one name if they are the same would help improve clarity.
- Throughout the paper the word "fewshot"/"fewshots" is used without a hyphen, which I'm pretty sure it should have one. For example on page 5, the phrase "10 fewshots" could either be changed to "10 few-shot examples", or "10 shots", but this occurs in many other places as well ("fewshot", "per-fewshot", etc.)

Minor Notes:
- There's a duplicated citation for Min et al.'s Rethinking the role of demonstrations: What makes in-context learning work?

___

**Generality of the findings**

Another concern I have is that even within the 5 tasks chosen, the proposed process by which the model performs few-shot ICL in each setting seems to be slightly different and task-specific. I'm not sure whether a sample size of 5 tasks is sufficient to claim generality that this two-step process (contextualization and aggregation) explains few-shot ICL in general, though it seems like a nice step. Studying the contextualization subcircuit in more tasks might help us understand the limits of these claims and would help to strengthen our confidence that this proposed framework is more general than the simple tasks presented.

___

**Missing Related Work**

One example of another position-aware circuit discovery algorithm is Haklay, et al. [3]. As you describing your own method, it would be helpful to know how your approach differs from theirs. The method by which you discover the position-level circuit not well-described. Particularly, how do you abstract away all edges into a single position? Is it by averaging or something else? Do you mainly treat all computation graph edges from one position to another as one "edge" in your position-level circuit, or is it multiple edges?

___
[1] Yin, et al. Which Attention Heads Matter for In-Context Learning? arxiv 2025. (https://arxiv.org/pdf/2502.14010)

[2] Lampinen, et al. The Broader Spectrum of In-Context Learning. arxiv 2024. (https://arxiv.org/pdf/2412.03782)

[3] Haklay, et al. Position-Aware Automatic Circuit Discovery. arxiv 2025. (https://arxiv.org/abs/2502.04577)

---

> ### Author Response · Authors · 2025-05-31
> **Comment by Authors - Part I**
>
> We are glad the reviewer found our work interesting and we thank the reviewer for all the constructive suggestions. We hope the following clarifications can address the reviewer's concerns.
>
> > The first is that the kind of in-context learning investigated in this paper is few-shot ICL prompting.
>
> Thank you for pointing this out, we will clarify this in introduction.
>
> > (1) The writing assumes familiarity with a lot of previous interpretability research.
> > (2) Giving a clearer definition of what you mean by "circuit" would probably be helpful.
> > (3) In section 3.1 "patching methodology", quite a few experimental details about the circuit-finding method and context are missing.
>
> We will add a paragraph in section 3.1 with information about patching setup and definitions, which we adopted from prior work. We provide the draft of the paragraph below:
>
> *To localize the behavior of the model, we use patching, an approach widely adopted in the literature (Wang et al., 2022; Hanna et al., 2023). With this technique, a model is viewed as a computation graph with activations in different layers as nodes, and computations between them as edges. Then, it is possible to ablate some of the edges in the graph, forcibly replacing the computation along a specific edge with the one computed on the counterfactual input. Counterfactual inputs are designed to erase relevant information from the prompt while leaving other information intact for even better localization of model behavior. For example, when substituting "Berlin" with counterfactual "Paris", we isolate computations specific to the city's identity, not those activated by its category (city) or token-type (word). If ablating a set of edges does not lead to a drop in the model's performance, then the information unique to the original input relative to the counterfactual input, which was transferred along these edges, is not causal for the model's prediction. This allows to discover the circuits - subgraphs of the full model's computation graph that can explain a major part of model's performance on a specific task.*
>
> *In contrast to the standard approach, we focus on the information flow between positions in a prompt; consequently, we differentiate between activations of the same heads at different positions.*
>
> For readers unfamiliar with mechanistic interpretability, we hope this section offers an accessible introduction to patching. We will provide an overview of patching approach, a detailed description of our experimental setup and its distinctions from standard methodologies in the Appendix. This will include basics of patching implementation, a thorough description of Figure 28, the metric we're measuring when doing patching between counterfactual and standard prompts, and the details of how we collapse multiple edges into one.
>
> >  Related to your specific setup, explaining in more detail how information across layers is treated for a single position would be helpful
>
> We will adjust the second paragraph in section 3.1 to include this information. Please see the draft below:
>
> *We first discover a *position-level circuit*. Nodes are the positions in the prompt; edges are directed arcs between them, representing the flow of information (Figure 1). Importantly, in position-level circuits we collapse all edges across layers into a single edge. Ablating such an edge means ablating all corresponding edges in every attention head and layer. For illustration, $x_2$ in a position-level circuit can only receive four incoming edges: from $x_1, t_1, y_1$, or $n_1$. Position-level circuits provide useful and interpretable upper bounds on information flow, though they do not distinguish between computations happening in different layers.*
>
> > (1) This submission does not have numbered lines
> > (2) The "parallel circuit" described in the paper appears to go by many names ("parallel circuit", "parallel subcircuit", "aggregation circuit", "aggregation-only circuit", etc.)
> >  (3) Throughout the paper the word "fewshot"/"fewshots" is used without a hyphen
> > (4) There's a duplicated citation for Min et al.'s Rethinking the role of demonstrations: What makes in-context learning work?
>
> Thank you for pointing this out. Indeed, we'll reduce references to "parallel circuit" to a single name, change the template, and adjust the text of the paper to include all of these changes.

---

> ### Author Response · Authors · 2025-05-31
> **Comment by Authors - Part II**
>
> > Another concern I have is that even within the 5 tasks chosen, the proposed process by which the model performs few-shot ICL in each setting seems to be slightly different and task-specific. I'm not sure whether a sample size of 5 tasks is sufficient to claim generality that this two-step process (contextualization and aggregation) explains few-shot ICL in general, though it seems like a nice step.
>
> We agree that this is a limitation of the paper, and that scaling the results to more difficult tasks is a nontrivial direction of future work. On the other hand, we also consider it a very nontrivial finding that some parts of the mechanism are shared between the tasks we used, particularly: the information that edges transport in different tasks, shared contextualization circuits between some of the tasks, cross-target-source edges ($x_i \rightarrow y_j$ and $y_i \rightarrow x_j$) being less important than other edges. Also, we believe it is expected that the exact mechanism is different for each task, and that putting them under one umbrella requires this umbrella to be quite abstract: which in our case is the concept of contextualization. Finding distinct smaller circuits for each task in our study goes somewhat beyond the abstraction, and this necessarily means that we identify more task-specific elements.
>
> > Missing Related Work:
> > One example of another position-aware circuit discovery algorithm is Haklay, et al. [3]. As you describing your own method, it would be helpful to know how your approach differs from theirs. The method by which you discover the position-level circuit not well-described.
>
> Thank you for pointing out this concurrent work, which indeed proposes a method quite similar to the one we're using. The differences can be summarized as follows: (1) the schema of the dataset in our case is well-defined by the tasks specifics: what we call $x$'s, $y$'s, $t$'s, $n$'s is called a span in Haklay, et al. For automated circuit discovery, we sum scores for the same edge between different tokens, and average over dataset examples, same as  Haklay, et al. (2) we define an edge from position A to position B as patching both V and K activations in position A when queried by position B, while Haklay, et al. differentiate between edges from Q, K and V activations. (3) We do not take edges from a position to itself into account and thus do not have separate nodes for MLPs, embeddings and logits in our graph. (4) In most of our experiments, we also abstract all edges between position A to position B in all layers into one edge to get a more interpretable mechanism. We differentiate between layers when finding the Activation-Level Circuits (Section 3.1).
>
> We will add detailed discussion in Appendix E.
>
> To specifically answer this question:
> > Particularly, how do you abstract away all edges into a single position? Is it by averaging or something else? Do you mainly treat all computation graph edges from one position to another as one "edge" in your position-level circuit, or is it multiple edges?
>
> We do not perform averaging within a position. Rather, as the reviewer suggests, we treat (in the position-level circuits) the set of all computation-graph edges from one position to another as a single edge of the position-level circuit. For position-level circuits, ablating a position-level edge from one position to another means ablating all computation-graph edges between this pair of positions; including the position-level edge means ablating none of these computation-graph edges. If *any* computation-graph edge plays an important role, we fully keep the overall position-level edge containing it. We will ensure that this is made explicit not just in the Appendix, but also in the main paper.

---

> ### Author Response · Authors · 2025-05-31
> **Comment by Authors - Part III**
>
> > On page 7, you mention that the few-shot ICL circuit is not a "classical induction circuit". While it may not copy literal tokens, do you think it could still be thought of as an "induction circuit" that "copies" forward a contextualized abstraction of the task?
>
> An induction-like mechanism might indeed still be happening, but our results suggest that it is not very selective to individual examples. Cho et al. argue that among the in-context examples the model chooses to copy only those that are most similar to the final query. We argue against this, as shuffling the K inputs in few-shot examples in these heads (which amounts to neutralizing attention weight differences between examples more or less similar to the query) does not affect model performance much. It may still be the case that $y_i\rightarrow t_{N+1}$ edges are a part of an induction-like circuit, but the sensitivity of this circuit to the specific query is not as strong as suggested by Cho et al. and various theoretical studies of ICL, as least in the model and tasks studied by us.
>
> > Do you think the phenomenon observed w/ person-sport has to do with the size of the output space?
>
> This is a very reasonable idea, and we hypothesize this phenomenon could indeed be attributed to the size of the output space and semantic meaningfulness of the labels. However, we did not experiment with this. Designing such an experiment is not so trivial, because one needs some care in finding tasks that have high accuracy in 2B model, and finding a set of suitable tasks might be tricky. However, this is an exciting direction for future work.
>
> >  ("The edges causally transport only task information", page 6) Which edges is this referring to? Is it all of the parallel-circuit edges ($x_i->y_i$, $x_{N+1}->t_{N+1}$, and $y_i->t_{N+1}$? Or only some subset of them?
>
> This refers to $y_i\rightarrow t_{N+1}$ only, we will clarify that in the text.
>
> > The current text relies too heavily on assuming the reader is familiarity with current interpretability research (particular circuit discovery methods) to understand it, and this undersells the nice findings presented in the paper. I suggest you try and present some more background on your experimental design and process to provide some intuition for those who may be less familiar with the process of "circuit-finding". I would also suggest to promote your "sub-findings" in section 3 (in boxes) earlier.
>
> Thank you for this suggestion. We will follow it, and spell out these findings already in the Introduction. We will also provide more background on the experimental design, especially in Section 3.1, to make the paper self-contained to a broader audience.

---

> > ### Comment · Reviewer_mZBA · 2025-06-02
> >
> > Thank you for the thorough reply, I think you've addressed all of my questions and cleared up some confusion I had regarding your circuit-finding method. I believe the changes you've proposed (including the text you've shared already) will strengthen the paper and hopefully make it more accessible to a broader audience. In light of these additions & clarifications I'm updating my score to Accept.

---

### Official Review · Reviewer_SjLE · 2025-05-10

**Rating:** 9
**Confidence:** 3
**Ethics Flag:** 1

**Summary:**

This paper studies in-context learning (ICL) for several tasks on Gemma-2 and argues that there is a two-stage flow of information: first, a contextualization circuit between input and output tokens of in-context examples build up representations, then an aggregation circuit connecting to the final separator token identifies the task to perform. This paper contributes to the growing body of work that seeks to understand ICL with mechanistic interpretability tools. While I do not fully follow all the details that support the claims (see below for questions/suggestions), I find the overall analysis and hypothesis to be thought-provoking.

**Questions To Authors:**

1. While the prose is generally clear, there are some details in the figures/tables that are difficult to discern. For example:
- Fig 2: What exactly is the difference between different version bars: do the blue and green bars correspond to parallel + contextualization? Is there aggregation involved? The caption can maybe be more descriptive.
- Fig 2: What does it mean when a circuit was chosen here? I understand the choice is based on accuracy but are you arguing there is a different "choice" for different tasks?
- Fig 3: Not significant - against what is this not significant?

2. Why do you call y_i->y_j / y_i -> y_{i+1} and x_i->x_{i+1} edges a "contextualization" subcircuit? I don't understand in what sense is this contextualizing. Do you mean it in the sense that higher layers of a token at position i can be viewed as a more contextualized word embedding of i (for example, word disambiguation)? Or do you mean contextualization as in the sense of in-context learning (but in that case I don't follow)?

3. It will be helpful to explain the methodology in more detail (I understand there is not much space so appendix is fine) For example, how about adding pseudocode for the patching methodologies in Section 3.1 and 3.2?

4. Potentially relevant citation (please ignore if not):

Where does In-context Learning Happen in Large Language Models? (NeurIPS'24)
https://proceedings.neurips.cc/paper_files/paper/2024/file/3979818cdc7bc8dbeec87170c11ee340-Paper-Conference.pdf

They show (via zeroing out attention on all x,y in-context tokens) that there is a critical set of middle layers in a LM that seem to do task recognition. I wonder if these results relate to your findings in any way.

5. Can you comment more on how focusing the five tasks (like Country-Capital) is a limitation (final paragraph of paper)?

**Reasons To Accept:**

Interesting and thought-provoking hypotheses about in-context learning

**Reasons To Reject:**

There is no major critique in particular for me, but I can imagine that some readers may not fully agree with the claims obtained by these kinds of mechanistic interpretability techniques.

---

> ### Author Response · Authors · 2025-05-31
>
> We are glad the reviewer found our work thought-provoking and we thank the reviewer for all the constructive suggestions. We hope the following clarifications can address the questions the reviewer raised.
>
> > some readers may not fully agree with the claims obtained by these kinds of mechanistic interpretability techniques
>
> While the methods we use are not perfect (we do not rule out the potential existence of multiple faithful, but distinct circuits, and our circuits recover close to, but not 100%, of the full model's performance), they are widely adopted in the literature and have become almost standard for this kind of work. Based on our experiments, we argue that the circuits we found are *causal* to the model's performance on the tasks, and we show causality via intervention in the model's behavior. Importantly, we avoid using *correlational* evidence, in order to make our analysis as rigorous as possible.
>
> > Fig 2: What exactly is the difference between different version bars: do the blue and green bars correspond to parallel + contextualization? Is there aggregation involved?
>
> These bars indeed correspond to parallel + contextualization; different colors correspond to different versions of the contextualization subcircuit. Aggregation in this context means the same as *parallel circuit*. We will update the legend to clarify that.
>
> > Fig 2: What does it mean when a circuit was chosen here?
>
> We test two versions of the contextualization subcircuit, which involve different edges between few-shot examples. In one version, all edges between ys are involved ($y_i \rightarrow y_j$); the other includes only edges between adjacent ys ($y_i \rightarrow y_{i+1}$) and also edges between adjacent xs ($x_i \rightarrow x_{i+1}$). In principle, we could have defined the contextualization subcircuit as the union of all those edges ($y_i \rightarrow y_j$ and $x_i \rightarrow x_{i+1}$), but we decided to use smaller subcircuits per task to make further analysis more tractable.
>
> As described in the paper, for each task, we chose the simplest circuit achieving ≥ 0.9 at N = 3, or (if there is none) the circuit achieving highest accuracy at N = 3, 10, based on 2B accuracies.
>
> > Fig 3: Not significant - against what is this not significant?
>
> This means that the drop in accuracy when doing a specific ablation is not significant according to a binomial test with α = 0.05 compared to the full circuit performance. We will clarify it in the caption.
>
> >  Why do you call $y_i\rightarrow y_j$ / $y_i\rightarrow y_{i+1}$ and $x_i\rightarrow x_{i+1}$ edges a "contextualization" subcircuit?
>
> We mean that the mid and higher layer representations of the i-th few-shot example become contextualized, so that they reflect certain information about the other (preceding) few-shot examples.
>
> For example, the higher layers of "Berlin" in a prompt "France\tParis\nBulgaria\tSofia\nGermany\tBerlin\nChina" in a contextualization circuit that includes edges $y_i\rightarrow y_{i+1}$, $x_i\rightarrow x_{i+1}$ can be contextualied with information from "Paris" or "Sofia", but not "France" or "Bulgaria". In a parallel circuit without contextualization, when the only edges we leave in a model are $y_i\rightarrow t_N$ and $x_i\rightarrow y_i$, higher layers of "Berlin" can not have any access to the information in previous few-shot examples (but it does have access to the information in embeddings of "Germany"), and thus its representaion is not contextualized with previous few-shot examples.
>
> >   It will be helpful to explain the methodology in more detail.
>
> We will extend section 3.1 to better describe the methodology we used, and also add a section in Appendix with more details about our setup and how it is different from standard patching. Apart from that, we will add a link to github with the implementation.
>
> Currently, the details can be found in Appendix D (definition of computation graph), Appendix G (counterfactual inputs) and Figure 28 (patching edges between positions).
>
> > Potentially relevant citation: Where does In-context Learning Happen in Large Language Models? (NeurIPS'24)
>
> Thanks for this pointer. We will add a citation and discussion. We believe that this finding is in good conceptual agreement with results about the role of function/task vector heads, which are in middle layers. Our work expands by clarifying the information flow leading up to the point where the task is recognized.
>
> > how focusing the five tasks is a limitation?
>
> Our analysis focuses on five tasks, and we cannot claim that our findings will directly generalize to more advanced tasks (math, reasoning, etc.) used in real applications. We limit the study to tasks on which the model has high accuracy and therefore do not discuss tasks with lower accuracy. Further modifications of the circuits might be needed to extend our analysis to other tasks.
>
> We thank the reviewer for all the suggestions and hope that additional clarifications make our findings more informative.

---

> > ### Comment · Reviewer_SjLE · 2025-06-05
> >
> > Thanks for the clarifications!

---

### Official Review · Reviewer_JCLW · 2025-05-11

**Rating:** 7
**Confidence:** 4
**Ethics Flag:** 1

**Summary:**

This work aims to identify the circuit for the formation of task vectors during in-context learning (ICL), focusing on 5 naturalistic tasks (namely, capitalization, country-capital, present-past, person-sport, copying) using the Gemma-2 2B model. Prior work has identified the presence of task vectors, i.e., outputs of some attention heads encode relevant information about the task which causally predicts the response. This work identifies a two-step mechanism using which the model infers the task information. They show that in lower layers, there is a parallel subcircuit where the model builds representation for each fewshot example (x_i, y_i) pair and also a contextualization mechanism where it connects the input and output tokens from all/some preceding examples. In the higher layers, the representations from the parallel subcircuit are aggregated for task identification and generating the prediction for the given input. They also show that the importance of the contextualize step can vary depending on the task, e.g. it can become more important in the presence of ambiguous examples.

**Questions To Authors:**

- Do the authors have ideas about what other models or tasks might have similar circuits for task vector formation in ICL? Similarly, it would be nice to discuss what tasks are significantly different yet amenable to mechanistic study where the circuit might be different, and which could be focus of future work.

- Other comments: The discussion on related work and the contextualization of this paper’s contribution compared to prior work is done well.

**Reasons To Accept:**

- This is a timely and rigorous mechanistic study on the formation of task vectors for ICL, which aids our understanding of this subject. The claims are well-supported by causal analysis: the paper shows that as seen in prior works, considering just aggregation only partially explains the model’s performance (e.g. it can explain about 85% of the model’s performance on 3/5 tasks, but <60% on capitalization and country-capital tasks), whereas considering both contextualization and aggregation explains at least 90% of the model’s performance on all 5 tasks.

- I like the section studying which information is routed, which goes beyond the evidence for circuit identification for information flow, and studies what (type of) information causally effects the downstream prediction.

**Reasons To Reject:**

Not reasons to reject but some things that the paper should address:
- The submission used the wrong template; there are no line numbers.
- The authors should explain the contextualization step in a bit more detail early on in the paper. For instance, in the abstract they discuss parallel subcircuit with contextualization, whereas in Fig. 1, the parallel subcircuit is used for the aggregation mechanism.
- Fig. 28 is helpful in understanding the patching methodology for circuit identification, so a small version could be included in the main text.

---

> ### Author Response · Authors · 2025-05-31
>
> We thank the reviewer for all the constructive suggestions and appreciate the acknowledgement of thoroughness and novelty of this work. We hope the following clarifications can address the questions the reviewer raised.
>
> > The submission used the wrong template
>
> Thank you for pointing at that, we will adjust the template accordingly.
>
> > The authors should explain the contextualization step in a bit more detail early on in the paper.
>
> Thank you for your suggestion. We will update the "Edges between different fewshot examples" paragraph on page 4 to include the definition of a contextualization subcircuit earlier.
>
> For reference, we include the draft of the change below:
>
> *Contextualization Subcircuit: Edges between different few-shot examples.* We next investigated which edges are needed beyond the Parallel subcircuit, aiming to find a small circuit recovering ≥ 90% of the accuracy of the full model. We call all the edges in this circuit except those in the Parallel subcircuit a Contextualization subcircuit. Note that this subcircuit can include slightly different edges for different tasks.
>
> > Fig. 28 is helpful in understanding the patching methodology for circuit identification, so a small version could be included in the main text.
>
> We will do our best to do this, the page limit permitting. We will extend section 3.1 with more details about the methodology we used, so that readers can refer to that early in the paper.
>
> > Do the authors have ideas about what other models or tasks might have similar circuits for task vector formation in ICL? Similarly, it would be nice to discuss what tasks are significantly different yet amenable to mechanistic study where the circuit might be different, and which could be focus of future work.
>
> The question of what task properties determine which circuit best explains the model's behavior is very interesting, and we currently do not have a rigorous answer for that. In preliminary experiments we found our contextualization circuits explain most of the model's behavior on tasks where the vanilla model achieves >90% accuracy. On the other hand, on tasks with lower accuracy (<75%) the circuits generally do not generalize well. We think the exact answer to this question is an exciting direction for future work.
>
> We thank the reviewer for all the questions and suggestions and hope that adding additional clarifications make our findings more informative.

---

> > ### Comment · Reviewer_JCLW · 2025-06-03
> >
> > Thank you for the response and clarifications.
> >
> > I maintain that this is an important and interesting work that advances our understanding of circuits formed during ICL in LLMs. While the paper focuses on one model and five tasks which can be viewed as a limitation (hence, not a higher score), the settings considered in the paper are investigated very thoroughly which makes it a solid work (hence, not a lower score). The paper poses interesting questions for future work and will be of interest to the broader LLM community. Therefore, I support acceptance of this paper.

---

### Official Review · Reviewer_cuaP · 2025-05-13

**Rating:** 5
**Confidence:** 2
**Ethics Flag:** 1

**Summary:**

The paper performs a mechanistic study of in-context learning: via interventions it finds a two-stage circuit that contextualizes
representations of individual examples, and then aggregates task information from them.  The study is performed on 5 simple and high-accuracy ICL tasks and the Gemma model family

**Reasons To Accept:**

- developing new insights on how LLMS perform ICL is still an important exciting problem
- mechanistic approaches and circuit design are a promising approach in this direction
- study seems to be rather detailed and contextualized with respect to related work

**Reasons To Reject:**

- unfortunately, my major concern about the paper is its lack of readability. I found the paper very very difficult to read and I identify two reasons for this: (i) continual references to (very recent) prior works Kharlapenko et al and Cho et al (ii) continual reference to figures in the appendix. These make reading very difficult and keep causing distractions. As someone who is high-level familiar with the circuit-design lit and mechanistic study of ICL, but is not familiar with these two prior works, I found myself unable to understand several parts of the authors claims and their descriptions of the methods. For context, the starting point of the section that I was hoping to get concrete details on the method begins with "We investigate circuits on the level of the individual positions in a prompt. Whereas Kharlapenko
et al. (2025) had collapsed different xi ’s into a single node (and same for all yi ’s), we keep them distinct, which allows us to understand information flow between fewshot examples."
- the study and its conclusions are limited to one model

---

> ### Author Response · Authors · 2025-05-31
>
> We are glad that the reviewer found our study to address an exciting problem, and to be detailed and contextualized with respect to related work.
>
> The reviewer notes as their *major concern* questions about readability. We are glad to address this as follows.
>
> > (i) continual references to (very recent) prior works
>
> We acknowledge that the paper has many references to this recent prior work. We did this in order to contextualize our work, and make the contributions transparent. We also are encouraged by the fact that the reviewer positively noted the contextualization of our work with respect to the literature.
>
> That said, in order to improve the flow for the reader, we will largely move comparison to Cho et al and Kharlapenko et al to a dedicated section in the Discussion.
>
> > (ii) continual reference to figures in the appendix
>
> We acknowledge that the paper has a good number of references to Appendix figures. We respectfully point out that the paper includes a large number of results, all of them quantitatively backed up with intervention results. We link our claims with tables and figures to show they are well-supported, but understanding the linked Appendix tables and figures is not necessary for following the main text. We note that it is infeasible to report the tables and figures backing up all of these in the main paper.
>
> We will add explicit notes, such as "(Appendix, Figure 19)" instead of "(Figure 19)". This will improve the flow for the reader, while maintaining easy access to the relevant figures and tables. We believe that this effectively addresses the reviewer's concern.
>
> > unable to understand several parts of the authors claims and their descriptions of the methods.
>
> To improve clarity in our description of the method, we will add a paragraph in section 3.1 with information about patching setup adopted from prior work and definitions. We will add a detailed description of our setup, and the way it is different from standard methodologies in Appendix. We provide the draft of the paragraph we will add to the main paper below:
>
> *To localize the behavior of the model, we use patching, an approach widely adopted in the literature (Wang et al., 2022; Hanna et al., 2023). With this technique, a model is viewed as a computation graph with activations in different layers as nodes, and computations between them as edges. Then, it is possible to ablate some of the edges in the graph, forcibly replacing the computation along a specific edge with the one computed on the counterfactual input. Counterfactual inputs are designed to erase relevant information from the prompt while leaving other information intact for even better localization of model behavior. For example, when substituting "Berlin" with counterfactual "Paris", we isolate computations specific to the city's identity, not those activated by its category (city) or token-type (word). If ablating a set of edges does not lead to a drop in the model's performance, then the information unique to the original input relative to the counterfactual input, which was transferred along these edges, is not causal for the model's prediction. This allows to discover the circuits - subgraphs of the full model's computation graph that can explain a major part of model's performance on a specific task.*
>
> *In contrast to the standard approach, we focus on the information flow between positions in a prompt; consequently, we differentiate between activations of the same heads at different positions.*
>
> > conclusions are limited to one model
>
> We would like to point out that we do have results on different models within the Gemma family. As described in the paper, our motivation for the Gemma model family is that it performs relatively better on relevant ICL tasks than other families even at the 2B scale. Thus, including other model families would necessitate studying them on different tasks, which makes the findings across models less comparable. That is why we decided to stick to one model family and leave others for future work. We also note that this approach is common in circuit discovery papers (Lindsey et al. [1], Wang et al. [2]), as it allows controlled in-depth study with substantial insights.
>
> [1] Lindsey, et al., "On the Biology of a Large Language Model", Transformer Circuits, 2025. (https://transformer-circuits.pub/2025/attribution-graphs/biology.html)
> [2] Wang, Kevin Ro, et al. "Interpretability in the Wild: a Circuit for Indirect Object Identification in GPT-2 Small." The Eleventh International Conference on Learning Representations. (https://arxiv.org/abs/2211.00593)
>
> Overall, we appreciate the reviewer's feedback on writing, and noting two specific directions. As the reviewer described readability as their *major concern*, we believe that the changes described above effectively address this concern, and hope that the reviewer reconsiders their score. If there is any further feedback on readability, we would be happy to address that, too.

---

> > ### Comment · Reviewer_cuaP · 2025-06-07
> >
> > Thank you for the response. I have read your response and also the rest of the reviews. I also read again your paper. I start by repeating the disclaimer that my confidence level reflects: while I am familar with ICL and have more closely followed its "theoretical" investigations (some of which you cite), I am not familiar with the mechanistic interpretability literature other than the induction-heads works. Thus, to me the process of circuit finding is not at all obvious and remains quite unclear to me even after the second read. On the other hand, after the second read I get a better picture of the claimed novelty of the findings over prior work and a (still vague) understanding of the messaging of the contextualization versus aggregation. My concerns about readability are reinforced by reading the rest of the reviews. I think it is clear that even reviewers that gave very high score seem to have had questions about the very basics of your methodology. I echo the suggestion given to you by Rev mZBA to please put effort in better explaining the meanings of concepts you use throughout "circuit finding", "aggregation", "contexutalization" (particularly, I found the meaning of "contextualization" very hard to pinpoint as there seems to be changing subcircuits given that name throughout without a unifying "definition" upfront). As someone that comes from possibly more theoretical background I actually find interesting the effort initiated in App A to link the results to synthetic ICL datasets like linear regression. Perhaps a more elaborate analysis in this "simpler" case (simpler in terms of task and in terms of model, e.g. 4 layer) could give transparency to your methodology and make it accessible to wider audience. Overall, the paper has high scores and, particularly as a non-expert, I have no reason to object to the paper getting accepted. With all good intentions, and an honest willingness to better understand your results and message, I will maintain a lower score for the reasons mentioned above. I would ideally also like to see some discussion about how can we use such information learnt from this study (this can be at the speculative level). Thank you

---

> > > ### Author Response · Authors · 2025-06-08
> > > **Part I**
> > >
> > > Thank you for the response and the effort invested in understanding our results. We're committed to improving the paper's readability and have already shared some text addressing concerns from you and other reviewers. We will address each of your concerns sequentially:
> > >
> > > 1. **Better explanation of patching methodology.**
> > >
> > > We will enhance readability for both experts and non-experts: Section 3.1 will retain its technical overview for experts, while gaining a new paragraph explaining core concepts in plain terms (the one we shared above). We will add a section describing our methodology in Appendix. This structure should enable general understanding while directing readers to the Appendix for in-depth explanations.
> > >
> > > We provide a draft of the section in Appendix below:
> > >
> > > *We use path patching  (Wang et al., 2022; Hanna et al., 2023) to identify position-level circuits and analyze information flow along edges. This method involves: (1) representing the model as a computation graph with token embeddings as inputs, loss value as output, and specific activations as nodes; (2) ablating ("removing") graph components to isolate those responsible for specific behaviors. This section details our computation graph definition (nodes, edges), loss function and ablation implementation.*
> > >
> > > **Computation graph**
> > >
> > > *Nodes consist of activations at the input/output of each attention head across positions. Each node is represented as a tuple (input/output, layer, head, position), where "position" indicates a token's role in the few-shot template (e.g., $x_1$, $y_3$, $t_2$, $n_4$). Note that $x_i$ and $y_i$ positions typically span multiple tokens but are treated as single nodes. Each node is thus a tensor of size (number of tokens in position, hidden dimension).*
> > >
> > > *Edges represent computational connections between nodes. In our position-level circuits, edges exist exclusively within attention heads, as only these transfer information between positions (Appendix E details edge inclusion criteria).*
> > >
> > > **Ablation Procedure**
> > >
> > > *Edge ablation requires two forward passes (Fig. 28):*
> > >
> > > 1. *Run the model on the counterfactual input $I_C$, saving all input node activations (input, L, H, pos)$_C$*. Here, the subscript $C$ denotes counterfactual activations.
> > > 2. *Run the model on the correct input $I$. For each attention head, compute outputs position-wise, which requires one forward passed through the attention block for each input position.
> > >  In order to ablate edge $\text{A} \rightarrow \text{B}$ in head H, layer L, for position B, we substitute the activation (input, L, H, A) at A with (input, L, H, A)$_C$ saved in Step 1. We concatenate the activations across positions to obtain the overall output of the attention head.
> > >  For instance, ablating an edge between (input, L, H, pos=2) and (output, L, H, pos=3) disconnects these nodes, implemented by computing:*
> > > (*output*,L,H,pos=3)=Attention((*input*,L,H,pos=1), (*input*,L,H,pos=2)$_C$, (*input*,L,H,pos=3))
> > > instead of
> > > (*output*,L,H,pos=3)=Attention((*input*,L,H,pos=1), (*input*,L,H,pos=2), (*input*,L,H,pos=3))
> > >
> > > *With this methodology we can identify circuits - subgraphs of the model's computation graph that can perform the task with high accuracy even when the edges outside the circuit are ablated.*
> > >
> > > **Ablations for Analyzing Information Routing**
> > > *For analyzing which information is routed through specific edge sets (Section 3.2), we partition edges into three categories:*
> > > (1) *Outside of the circuit,*
> > > (2) *Inside the circuit but not under investigation,*
> > > (3) *Inside the circuit and under investigation.*
> > > *We run the following extended procedure:*
> > >
> > > 1. *Run counterfactual input $I_C$, saving (input, L, H, pos)$_C$*
> > > 2. *Run semi-counterfactual input $I_{SC}$, ablating edges outside circuit (Set 1). Save input node activations for heads corresponding to Set 3 edges: (input, L, H, pos)$_{SC}$*
> > > 3. *Run correct input $I$ while ablating:*
> > >     - *Set 1 edges using (input, L, H, pos)$_C$*
> > >     - *Set 3 edges using (input, L, H, pos)$_{SC}$*
> > >
> > > *This isolates performance changes caused by removing only Set 3 information not present in the semi-counterfactual. Counterfactual inputs (e.g., replacing "Berlin" with "table") preserve less information than semi-counterfactual (e.g., replacing "Berlin" with "Paris"), which preserve broader categorical attributes. This reveals which information passed along specific edges impacts downstream performance.*
> > >
> > > **Handling Multiple Tokens per Position**
> > >
> > > *Our approach requires identical token counts per position in counterfactual and correct prompts, enforced during construction. When ablating edges from position A, we replace its entire activation tensor with the counterfactual version.*
> > >
> > > *For projecting position-level circuits to activation-level circuits, we assign each edge a single importance score by summing scores across all constituent token-level edges. Position-level circuits require no such aggregation. See Appendix F for activation-level circuit details.*

---

> > > ### Author Response · Authors · 2025-06-08
> > > **Part II**
> > >
> > > 2. **Clarifying what "contextualization" means.**
> > >
> > > We appreciate Reviewer cuaP's and JCLW's feedback on this term and have refined its definition for greater clarity. We will also add a formal mathematical specification of what we mean by contextualization. We provide the revised paragraph below:
> > >
> > > *We next investigated which edges are needed beyond the Parallel subcircuit, aiming to find a small circuit recovering $\geq 90\%$ of the accuracy of the full model. We call the set of edges in this circuit -- except those already appearing in the Parallel subcircuit -- a Contextualiztion subcircuit. Specifically, a Contextualiztion subcircuit comprises edges that equip the representation of $(x_i, y_i)$ with information about prior few-shot examples $(x_j, y_j)$, where $j < i$. This is in contrast to the Parallel subcircuit, where the representation of $(x_i, y_i)$ receives no information about other few-shot examples. The edges in a Contextualization subcircuit connect nodes $A \rightarrow B$, with $A \in \{x_j, y_j | j < i\}$ and $B \in \{x_i, y_i\}$ for $i \leq N + 1$. Note that different subsets of these possible edges might be needed across tasks.*
> > >
> > > 3. **How can we use information learnt from this study.**
> > >
> > > We believe that studying ICL circuits is useful for two reasons. First, practically: ICL is a popular use case of LLMs, and understanding how it works helps us get more robust responses and make model outputs more trustworthy. Second, scientifically: ICL is an emergent ability that only appears in larger models. Understanding how it works gives us clues about why scale changes model behavior—which is a major open question in the field.
> > >
> > > Even though we conducted the study only on simpler tasks (where the model gets high accuracy), we still gained new insights:
> > > 1. We show that contextualization matters (unlike what some earlier work suggested [1]).
> > > 2. We show that  contextualization edges play an incresingly important role when individual few-shot examples are ambiguous.
> > > 3. We identify the information trasferred along each set of edges, which (1) might explain prior empirical findings, e.g. Min et al. [2] ’s finding that sometimes just seeing the output space is enough for the model to perform the task; (2) shows that contextualization edges do not pass exact token details, only "type" info (like "city" or "country").
> > >
> > > We believe that, in this study, we provide interesting empirical observations about ICL that were not documented before, and open up many directions for future work, including scaling results to other model families and tasks, and asking deeper questions as to "why" this happens.
> > >
> > > We’ll add these takeaways to the Discussion section.
> > >
> > > We thank the reviewer for the suggestions to improve the text and hope the shared Appendix section clarifies both our methodology and planned revisions.
> > >
> > > [1] Cho, Hakaze, et al. "Revisiting in-context learning inference circuit in large language models." arXiv preprint arXiv:2410.04468 (2024).
> > >
> > > [2] Min, Sewon, et al. "Rethinking the Role of Demonstrations: What Makes In-Context Learning Work?." Proceedings of the 2022 Conference on Empirical Methods in Natural Language Processing. 2022.

---

### Decision · Program_Chairs · 2025-07-08

**Decision:**

Accept

**Comment:**

The paper investigates how a Gemma-2 model pieces together task information from few-shot prompts. Using causal interventions, the authors identify a two-stage flow: lower-layer “contextualization” edges interleave each (x , y) example with its predecessors, while higher-layer “aggregation” edges pool those contextualized representations to form the task vector and produce the answer. Experiments on five naturalistic tasks and additional synthetic regression show that combining the two circuits accounts for a large fraction of model accuracy, whereas aggregation alone can fall short on the harder tasks.

Reviewers found the study timely and well-situated in the mechanistic-interpretability literature. They praised the rigorous intervention methodology, the clear empirical evidence that contextualization matters when examples are ambiguous, and the thoughtful comparison with recent circuit-finding work. The principal reservations centered on presentation: the original draft leaned heavily on recent papers, made frequent trips to appendix figures, and assumed readers already understood patching techniques. Further concerns involved the narrow focus on a single model family and the use of a non-standard template.

The author response addressed these points in depth. New text will define “circuit,” “contextualization,” and the patching protocol up front, include a schematic of Figure 28 in the main body, and standardize terminology. The authors also pledged to move comparisons with Cho et al. and Kharlapenko et al. to a dedicated related-work section, correct the template issues, and add discussion of how the findings could inform future work on larger tasks or other model families. Reviewers indicated that these revisions resolve their substantive concerns, leaving mostly editorial polish. Should the paper advance, it would benefit from integrating the new methodological paragraph into Section 3 and expanding the appendix description of position-level vs. activation-level edges.